# Genomic insights into rapid speciation within the world's largest tree genus *Syzygium*

Yee Wen Low [1,2,3,33] ✉, Sitaram Rajaraman [4,5,33], Crystal M. Tomlin[6,33], Joffre Ali Ahmad[7], Wisnu H. Ardi[8], Kate Armstrong [9], Parusuraman Athen[1], Ahmad Berhaman[10], Ruth E. Bone[2], Martin Cheek[2], Nicholas R. W. Cho[4], Le Min Choo[1], Ian D. Cowie[11], Darren Crayn [12], Steven J. Fleck[6], Andrew J. Ford[13], Paul I. Forster[14], Deden Girmansyah[15], David J. Goyder [2], Bruce Gray[12], Charlie D. Heatubun[2,16,17], Ali Ibrahim[1], Bazilah Ibrahim[1], Himesh D. Jayasinghe [18,19], Muhammad Ariffin Kalat[7], Hashendra S. Kathriarachchi[18], Endang Kintamani [15], Sin Lan Koh[1], Joseph T. K. Lai[20], Serena M. L. Lee[1], Paul K. F. Leong[1], Wei Hao Lim[1], Shawn K. Y. Lum[21], Ridha Mahyuni[15], William J. F. McDonald[14], Faizah Metali[22], Wendy A. Mustaqim [23], Akiyo Naiki [24], Kang Min Ngo[21], Matti Niissalo[1], Subhani Ranasinghe[25], Rimi Repin[26], Himmah Rustiami[15], Victor I. Simbiak[17], Rahayu S. Sukri[22], Siti Sunarti[15], Liam A. Trethowan[2], Anna Trias-Blasi[2], Thais N. C. Vasconcelos[2,27], Jimmy F. Wanma[17], Pudji Widodo[28], Douglas Siril A. Wijesundara [19], Stuart Worboys [12], Jing Wei Yap[29], Kien Thai Yong[30], Gillian S. W. Khew [1,32], Jarkko Salojärvi [4,5], Todd P. Michael [31], David J. Middleton [1], David F. R. P. Burslem [3], Charlotte Lindqvist [4,6] ✉, Eve J. Lucas [2] ✉ & Victor A. Albert [4,6] ✉

Species radiations, despite immense phenotypic variation, can be difficult to resolve phylogenetically when genetic change poorly matches the rapidity of diversification. Genomic potential furnished by palaeopolyploidy, and relative roles for adaptation, random drift and hybridisation in the apportionment of genetic variation, remain poorly understood factors. Here, we study these aspects in a model radiation, *Syzygium*, the most species-rich tree genus worldwide. Genomes of 182 distinct species and 58 unidentified taxa are compared against a chromosome-level reference genome of the sea apple, *Syzygium grande*. We show that while *Syzygium* shares an ancient genome doubling event with other Myrtales, little evidence exists for recent polyploidy events. Phylogenomics confirms that *Syzygium* originated in Australia-New Guinea and diversified in multiple migrations, eastward to the Pacific and westward to India and Africa, in bursts of speciation visible as poorly resolved branches on phylogenies. Furthermore, some sublineages demonstrate genomic clines that recapitulate cladogenetic events, suggesting that stepwise geographic speciation, a neutral process, has been important in *Syzygium* diversification.

Species radiations—wherein perplexing amounts of diversity appear to have formed extremely rapidly—have featured prominently in the history of evolutionary theory[1]. Various underlying mechanisms for their formation have been proposed[2], including adaptation[2], non-adaptive processes[3,4], hybridisation[5,6], and polyploidy[7,8], but the relative importance of these drivers remains incompletely understood. Species radiations on islands have been among the most prominently studied systems[9–12]. For example, the Malesian archipelago in the tropical Far East[13], consisting of thousands of islands and including New Guinea and Borneo, the second and third largest islands in the world, is a biodiversity hotspot containing many radiations of plant and animal species. Among forest trees, local tree species richness across Southeast Asian forests is largely driven by a small number of highly species-rich genera[14]. The clove genus, *Syzygium*, is one of the most important of these genera, and therefore understanding diversification and its underlying drivers within *Syzygium* may help explain large-scale patterns of diversity in the Palaeotropics. However, *Syzygium*, like many other species radiations that hold immense morphological and ecological variation, has so far been difficult to resolve phylogenetically[15–18], leading to the impression that evolutionary change can be a swift process that may not require substantial underlying genetic change[9]. Here, we employ genome-scale approaches to investigate speciation patterns and their potential drivers in the most species-rich tree genus worldwide, *Syzygium*[19].

*Syzygium*, which includes 1193 species recognised worldwide[20], is a genus in the myrtle family (Myrtaceae). *Syzygium* is restricted to tropical and subtropical regions of the Old World, where it is distributed from Africa through to India, across Southeast Asia and extending to Hawaii in the Pacific Ocean, with the centre of species diversity in Indomalesia[20]. The type species of *Syzygium* is *S. caryophyllatum*, a poorly known, small to medium-sized tree endemic to southern India and Sri Lanka[21]. The best-known species in the genus is the clove tree, *Syzygium aromaticum*, from which flower buds are gathered, dried, and used as a spice, a preservative and in pharmacology[22]. In addition, *Syzygium aqueum*, *S. cumini*, *S. jambos*, *S. malaccense* and *S. samarangense* are widely cultivated in the tropics for their large edible fruits[23]. *Syzygium samarangense* is cultivated commercially in Southeast Asia, where it is marketed as the wax apple, java apple, rose apple, or samarang rose apple. Apart from being used as cooking ingredients or cultivated for fruits, *Syzygium* species with dense and bushy crowns, such as *S. antisepticum*, *S. australe*, *S. luehmannii*, *S. myrtifolium* and *S. zeylanicum*, are used in the horticulture industry in Australia, Indonesia, Malaysia and Singapore for hedges, natural fences, natural sound barriers and privacy screens[24].

*Syzygium* species are generally medium-sized to large, characteristically sub-canopy trees that are sometimes emergent, while some also form shrubs, small forest understorey treelets, swamp and mangrove forest trees, and rheophytic vegetation[25]. As is true of many tropical trees, *Syzygium* flowers are visited by a large diversity of insects and vertebrates, and their fruits are typically eaten by a variety of flying and arboreal vertebrates and even terrestrial bird, mammal and reptile browsers[25]. *Syzygium* species also occur as dominant mid-level canopy trees, affecting the ecosystems of plants, animals, and fungi in lower forest layers[25]. Many species co-occur; for example, there exist ca. 50 taxa on a single 52-ha. ecological plot in the Lambir Hills National Park (Sarawak, East Malaysia, Borneo[26]), where they display fine-scale differentiation in habitat occupancy and stature[14]. The genus is notorious as one of the most difficult to identify due to the paucity of clear, diagnostic morphological characters for distinguishing species;[25,27,28] morphological variation in the genus can appear as continua of traits rather than collections of discrete units. Given the immense number of species assigned to *Syzygium*, it contributes disproportionately to the diversity of Southeast Asian tropical

forests. Therefore, understanding diversification and its underlying drivers within *Syzygium* may help explain large-scale patterns of diversity. Thus far, however, phylogenetic studies of *Syzygium* have involved only a few PCR-amplified plastid and nuclear marker genes[15,16]. An infrageneric classification proposed in 2010 was based on three plastid loci[17], and although it resolved some major clades, interrelationships within the bulk of the genus, species of *Syzygium* subg. *Syzygium*, were left largely unresolved.

Here, we sequence whole genomes to vastly increase the available data in an attempt to more fully resolve phylogenetic relationships among *Syzygium* species. We use Oxford Nanopore Technology (ONT) long-read sequencing[29] to assemble and annotate a chromosome-scale reference genome for the sea apple, *Syzygium grande*[23]. This species was selected as a representative because it is a well-known member of the most diverse, broadly distributed group within *Syzygium*, and one of the most commonly cultivated shade and firebreak trees planted along streets in Singapore and Peninsular Malaysia. We examine the palaeopolyploid history of *Syzygium* to assess whether whole genome duplications may have played a role in speciation through sub- or neo-functionalisation events, eventually fixed by natural selection or drift processes during species transitions[7]. We use whole-genome sequencing of 292 *Syzygium* individuals and outgroups to address evolutionary relationships among the species. Both Illumina short-read assemblies, as well as mapping of the read data to the *S. grande* genome, are brought to bear for phylogenomic investigations of possible rapid diversification in the group.

## Results and discussion

### Assembly and annotation of the reference and resequenced *Syzygium* genomes

A chromosome-level assembly of *Syzygium grande* (Fig. 1a) was carried out using wtdbg2[30] to generate a 405,179,882 bp genome in 174 contigs (N50 of 39,560,356 bp) from more than 60 Gb of ONT long reads, and the assembly was subsequently polished with 30 Gb Illumina short reads. Finally, scaffolding into pseudo-chromosomes was carried out using Dovetail HiC technology[31] to generate 11 pseudo-chromosomes (Fig. 1b).

Following the assembly, repeat masking (Supplementary Table 1) and gene prediction were carried out using evidence from *Syzygium grande* RNA-seq data and protein sequences from *Arabidopsis thaliana* and *Populus trichocarpa*. Altogether 39,903 gene models were predicted with 86.6% of benchmarked universal single-copy orthologous (BUSCO 3.0.2[32]) genes being present. In addition to the reference assembly, 30 Gb of Illumina HiSeqX sequencing data was generated for each of 289 *Syzygium* individuals and three outgroup taxa (two *Metrosideros* and one *Eugenia* species, both Myrtaceae; Supplementary Data 1) and assembled de novo using the MaSuRCA assembler[33] (Supplementary Data 2 and Supplementary Fig. 1). The average single-copy completeness across this set of genomes was 89.23% (Supplementary Data 2), indicating that the draft assemblies were of acceptable quality.

### Genome structure of *Syzygium* reveals that a single polyploidy event underlies all Myrtales

We used our chromosome-level assembly of *Syzygium grande* to re-evaluate the polyploid history of its family, Myrtaceae, and order, Myrtales. Myrtales are a diverse rosid lineage comprising approximately 13,000 species across 380 genera and 9 families[34]. All rosids share the *gamma* triplication event that occurred in the core eudicot common ancestor[35–37]. Sequencing of the *Eucalyptus grandis* (Myrtaceae) genome revealed an additional whole genome duplication (WGD) in its lineage[38], and later analyses of the *Punica granatum* (pomegranate) genome in the related family Lythraceae suggested that this polyploidy event may have been shared[39,40], occurring near the base of the order. Further work on the *Psidium guajava* (guava)

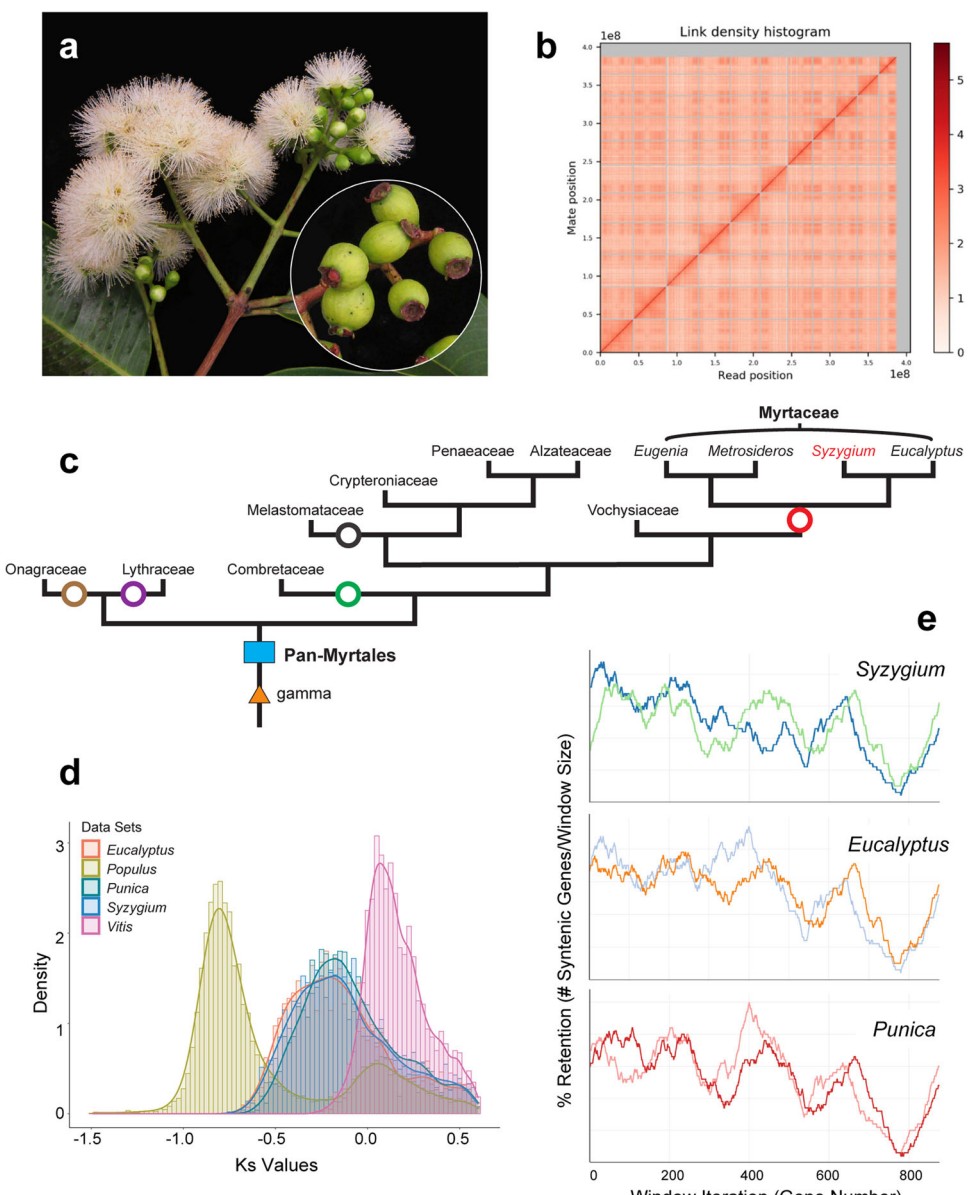

**Fig. 1 | Assembly and structural evolution of the *Syzygium grande* reference genome. a** *S. grande* inflorescence, flowers and fruits; the latter evoke the common name "sea apple"; **b** HiC contact map for the scaffolded genome, showing 11 assembled chromosomes; **c** Phylogeny of major lineages of Myrtales, following Maurin et al. [44]. Genera of Myrtaceae used in genome structural and phylogenetic analyses are also depicted. *Punica* (Lythraceae) was also examined for structural evolution. Open circles represent the multiple, independent polyploidy events predicted by the 1KP study[42]; our results here suggest instead a single Pan-Myrtales whole genome duplication (blue rectangle) which followed the gamma hexaploidy (orange triangle) present in all core eudicots. **d** Synonymous substitution rate density plots for internal polyploid paralogs within *Syzygium, Eucalyptus, Punica,* *Populus* and *Vitis*. Modal peaks in these three Myrtales species suggest a single underlying polyploidy event. Ks asymmetries were calibrated using the *gamma* event present in each species. Both histograms and smoothed curves are shown. **e** Fractionation bias mappings of Myrtales chromosomal scaffolds, 2 each (different colours), onto *Vitis vinifera* chromosome 2 show similar patterns for all three Myrtales species (excluding cases of chromosomal rearrangements among the three, which are discernible as different scaffold colour switchings compared to the *Vitis* chromosome). *X*-axis shows the percent retention of fractionated gene pairs following polyploidisation; *Y*-axis shows the position of gene pairs along the *Vitis* chromosome. Photograph credit: WHL (**a**). Source data are provided as a Source Data file.

genome came to a similar conclusion[41]. However, the broad, transcriptome-based 1KP project suggested that the Lythraceae and Myrtaceae WGDs might be independent events. Indeed, seven independent, lineage-specific WGDs were predicted by 1KP (their Supplementary Fig. 8) to characterise a larger lineage containing *Larrea, Tribulus* (both Zygophyllaceae), Combretaceae, Onagraceae, Melastomataceae, Lythraceae and Myrtaceae[42].

Syntenic alignments of the *Syzygium grande* genome against itself revealed at least one whole genome multiplication event since the *gamma* palaeohexaploidy (Supplementary Fig. 2), and alignment

against the *Vitis vinifera* genome confirmed the single lineage-specific WGD (Supplementary Fig. 3). A more detailed study against both *Eucalyptus grandis* and *Punica granatum* revealed 1:1 syntenic relationships (Supplementary Figs. 4 and 5), strongly suggesting a shared polyploid history. We investigated this further by extracting internally syntenic gene pairs in *Eucalyptus grandis, Punica granatum, Vitis vinifera* and *Populus trichocarpa*. When rate-corrected against the *gamma* hexaploidy event[43], an ancient pan-Myrtales WGD was supported, approaching *gamma* in age (Fig. 1c and d). Furthermore, subgenome-wise syntenic depths and fractionation patterns were extremely similar

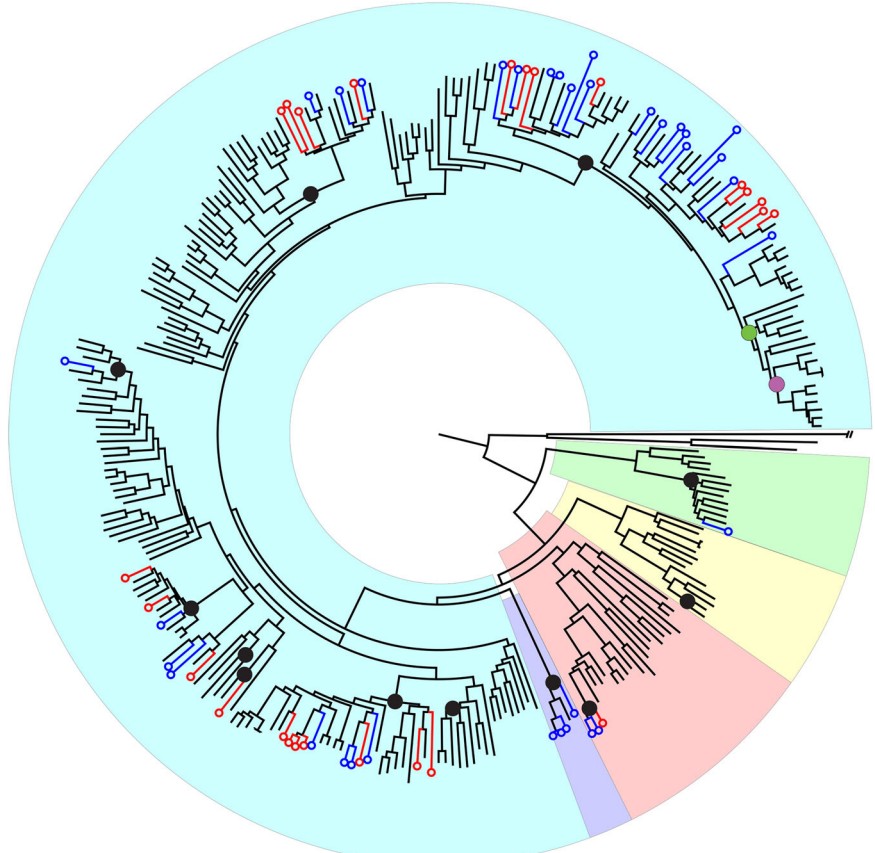

**Fig. 2 | Phylogenetic tree based on single nucleotide polymorphisms among all 292 resequenced Myrtaceae accessions.** Black circles represent the at-least 12 independent invasions of Sunda from Sahul. The green circle represents migration from Sunda to the Indian subcontinent, and the purple circle denotes further migration from there to Africa. Blue versus red circles at the leaves of the tree represent *Syzygium* accessions from Bukit Timah Nature Reserve and Danum Valley Conservation area, respectively. Background colours represent the recognised subgenera, (clockwise from the root, excluding the outgroup taxa) *Syzygium* subg. *Sequestratum* (green), *S.* subg. *Perikion* (yellow), *S.* subg. *Acmena* (red), the *S. rugosum* clade (purple) and *S.* subg. *Syzygium* (cyan).

in *Syzygium grande*, *Eucalyptus grandis*, and *Punica granatum*, supporting the hypothesis that a single polyploidy event underlies all Myrtales (Fig. 1e and Supplementary Fig. 6). Furthermore, alignment of *Populus trichocarpa* against given *Syzygium grande* chromosomes showed the expected 2:1 syntenic pattern indicative of an independent Salicaceae-specific WGD in the rosid order Malpighiales (Supplementary Fig. 7). Based on phylogenetic relationships recently solidified for Myrtales families[44], we conclude that earlier genome-based determinations of shared polyploid status within Myrtales are correct in indicating one basal WGD, and that the transcriptome-based 1KP study erroneously inflated the number of WGDs within the clade (Fig. 1c).

Since some polyploid events such as the *gamma* triplication[35–37] and the pan-angiosperm WGD[45] co-occur with major flowering plant radiations[7] (here, the core eudicots and all angiosperms, respectively), a single polyploid event shared by all Myrtales might hold implications for early diversification in the order. However, it is well-known that some large angiosperm diversifications, such as Gentianales (an even larger lineage than Myrtales at >20,000 species across 1121 genera and 5 families[34]), are not marked by ancestral WGDs, leaving polyploidy as a causal mechanism for diversification rather inconclusive, or at the very least an incomplete explanation.

Polyploidy within *Syzygium* similarly appears to play little role in its infrageneric diversification. BUSCO duplicate (*D*) scores suggest that the majority of species have remained at the same ploidy level following the Pan-Myrtales WGD event (Supplementary Data 2). At least one clear case of neopolyploidy is observable in *S. cumini*, which has the highest *D* score in our sample and a known haploid chromosome number of $n = 22$[46], double the number of our *S. grande* pseudochromosomes.

## Single nucleotide polymorphisms and single-copy nuclear genes yield well resolved major branchings within *Syzygium*

Species-level interrelationships within *Syzygium* have not yet been investigated in depth. To obtain a whole genome-level phylogeny we used the *Syzygium grande* genome assembly as a reference for mapping variants from three outgroup taxa and 289 independently sequenced *Syzygium* accessions representing at least 182 distinct species, 49 repeated species samples, and 58 additional as-yet-unidentified taxa. SNP calling yielded 1,867,173 variants across all 292 samples, from which we determined genome-wide phylogenetic relationships using RAxML[47] (Fig. 2). Since the SNPs could be identified only from the relatively conserved parts of the genome, we also collected predicted universal single-copy genes from BUSCO analyses (source data are provided at Dryad, https://doi.org/10.5061/dryad.h18931zpw) estimate using ASTRAL[48], a coalescence-based approach that incorporates individual gene trees into species tree estimation.

Phylogenetic analysis using genome-wide SNPs (source data are provided at Dryad, https://doi.org/10.5061/dryad.h18931zpw) resulted in a phylogeny (Fig. 2 and Supplementary Fig. 8) that was robust and well-resolved with the outgroup-based rooting, namely *Metrosideros excelsa*, *M. nervulosa* (tribe Metrosidereae) and *Eugenia reinwardtiana* (tribe Myrteae). Five major clades resolved in the phylogeny were all well-supported, with most branches receiving 100% bootstrap support at the nodes, indicating strong internal consistency within the dataset. These five major clades represent the previously characterised

*Syzygium* subg. *Syzygium*, *S.* subg. *Acmena*, *S.* subg. *Perikion*, *S.* subg. *Sequestratum* and a subgenus yet to be named that includes *S.* cf. *attenuatum*, *S. rugosum* and an unidentified species from Sulawesi labelled here as "SULAWESI2" (henceforth, we refer to this lineage as the *S. rugosum* clade). It is noteworthy that internal branch lengths are heterogeneous in length, indicating that the clades are differentially divergent either in time, diversification rate, population size, or all of these factors[49]. The largest clade in the phylogeny, having both the most recognised species and the most representative individuals in our current sample, is the *Syzygium* subg. *Syzygium* clade. Relationships within this clade are well resolved and supported, as are interrelationships among the five subgenera. Despite strong support, it is important to note that such an analysis generates a phylogeny that represents a genome-wide average, rather than taking into account the independent inheritance of different loci across the genome characteristic of incomplete lineage sorting or adaptive processes.

To obtain an independent view of the *Syzygium* species tree, we used our BUSCO single-copy gene sets (source data are provided at Dryad, https://doi.org/10.5061/dryad.h18931zpw) to compare the gene trees derived from independent nuclear loci in a coalescence species tree approach. We analysed two different BUSCO gene sets that differed in their completeness among accessions: the set of 229 genes containing representatives from all sequenced individuals, and a second set with 1227 genes present in ~95% of accessions. The phylogenies obtained from both single-copy genes and genome-wide SNPs (Supplementary Fig. 8) concordantly displayed the five major, well-supported clades representing the five subgenera of *Syzygium*, including also their relative branching order from the outgroup root, albeit with some minor disagreement of taxon placement within clades. These corroborating results inferred from two different approaches indicate strong and consistent phylogenetic signals within our genomes. Furthermore, *Syzygium* interrelationships based on plastid mappings (source data are provided at Dryad, https://doi.org/10.5061/dryad.h18931zpw), derived by mapping the Illumina sequence reads for each accession onto the *Syzygium grande* plastid genome, yielded partly incongruent results that may be traceable to ancient hybridisation and plastome capture, or to incomplete lineage sorting (ILS) (Supplementary Fig. 9).

## Diversification bursts characterise many terminal branchings within *Syzygium* phylogeny

Despite a strong overall signal supporting a bifurcating evolutionary history, the many extremely short coalescent branch lengths generated by the ASTRAL approach suggest that ILS[49] may have been a confounding biological factor at various points during the *Syzygium* radiation. These branch lengths, which are interpretable in terms of time in generations (*g*) divided by effective population size ($N_e$)[49], provide evidence that many *Syzygium* clades either radiated extremely rapidly, or that their ancestral population sizes were comparatively large, or both. Such *g* and $N_e$ conditions are known to promote gene-tree/species-tree discordance through ILS[49]. We sought to investigate signatures of ILS in the data further using NeighborNet, a distance-based method based on neighbour-joining that generates phylogenetic networks[50]. The character incongruence that is manifested as extra edges in these networks beyond a perfectly bifurcating tree has been interpreted both in terms of interspecies admixture and/or incomplete lineage sorting phenomena[51,52].

NeighborNet analysis of our genome-wide SNP data for *Syzygium* subg. *Syzygium* including a single outgroup species, *S. rugosum*, showed that while many of the evolutionary relationships among taxa were strongly tree-like, at least one major clade (which we informally term here the "*Syzygium grande* group") likely involved a burst of lineage splits (Fig. 3a), as evidenced by the predominantly noncoding (i.e., neutrally evolving) SNPs which illustrate a highly webbed, fan-like network of splits at its base. While the stem lineage of the *Syzygium*

*grande* group was strongly supported in the BUSCO and SNP trees, it is noteworthy that in the SNP analysis many parallel edges nonetheless appear along it, suggesting internal incongruence among SNPs, possibly reflecting differential inheritance with ILS (see Suh et al.[52]). A further, larger lineage including the *Syzygium grande* group and its outgroups was similarly well supported in the BUSCO and SNP trees; however, its own stem lineage contained even more parallel edges and potentially even more severe ILS (Fig. 3a).

## Incomplete lineage sorting rather than hybridisation may confound phylogenetic inferences

Next, we used the same SNP data with the ADMIXTURE software[53] to search for genomic partitioning among the clades and accessions that might be attributable to admixture (introgression) or differential blockwise inheritance through extremely narrow species splits (ILS). ADMIXTURE assumes *K* ancestral population clusters on the data; it is not decisive regarding mechanisms underlying any *K*-cluster mixtures within individuals analysed. The approach was developed for population-level data wherein mixed *K*-clusters are most likely attributable to admixture rather than ILS through lineage splits (e.g., speciation events). However, results at the interspecific level are often interpreted uncritically as actually indicative of cross-lineage admixture[54] (see ref. [55]). Indeed, the *K* components from ADMIXTURE simply represent subsets of inherited SNP variation that could reflect any underlying mixtures, of which ILS can be one mechanistic basis (Supplementary Figs. 10 and 11). We, therefore, propose ILS to be a likely underlying causal factor for some of the *K* mixtures given both the short coalescence branch lengths on the ASTRAL species tree and the reticulation of the NeighborNet.

Our ADMIXTURE analysis cross-validation scores supported *K* = 14 as the best representation of ancestral population structure (Supplementary Fig. 12). At this *K*, the *Syzygium grande* group is almost entirely assigned to one *K* in the ADMIXTURE results (orange, Supplementary Fig. 11). However, at other *K* values (e.g., *K* = 10,11,12; Supplementary Figs. 10 and 11), *K* mixtures within this group are apparent. It is worth noting that the fold level of cross-validation affects the preferred number of components, since the optimum results in a case where for each component there is at least one representative in the test set. The outgroups to the *Syzygium grande* group also contain the orange-coloured cluster at *K* = 14, but they additionally include mixtures with other ancestral populations (Supplementary Fig. 11). These *K*-cluster mixtures appear to be consistent with the multiple edges underlying this larger lineage in the NeighborNet analysis. In other words, they are likely indicative of differential inheritance of genomic regions and their SNPs through ILS. To rule out admixture generating these results, we formally tested for gene flow within the *Syzygium grande* group using Patterson's $f_3$ statistic[56], which tests for patterns of allele sharing (source data are provided at Dryad, https://doi.org/10.5061/dryad.h18931zpw). We calculated all three-way taxon comparisons of source1, source2, and target taxa to evaluate signatures of admixture. These results demonstrated no evidence for admixture, but did reveal instances where significant negative *Z*-scores across all possible source combinations reflected close relationships through identity by descent (described in detail by Lan et al.[57]) (Supplementary Figs. 13–87).

With ILS the more likely explanation for these results, we used local principal component analysis (PCA)[58] to examine whether patterns of SNP-based relatedness differed instead by location along chromosomes. Clear distinctions in the sample projections on PCA components along a scaffold would indicate that different genomic blocks have different evolutionary histories, of which introgression, ILS, local selection, or even drift are suspect source mechanisms. Local PCA takes window-wise PCA projections of SNP variation and arrays differences among them on a multidimensional scaling (MDS) plot; three distinct "corners" are then selected from the MDS plot, and the corner-wise variation is pooled for final analyses[58]. We analysed both

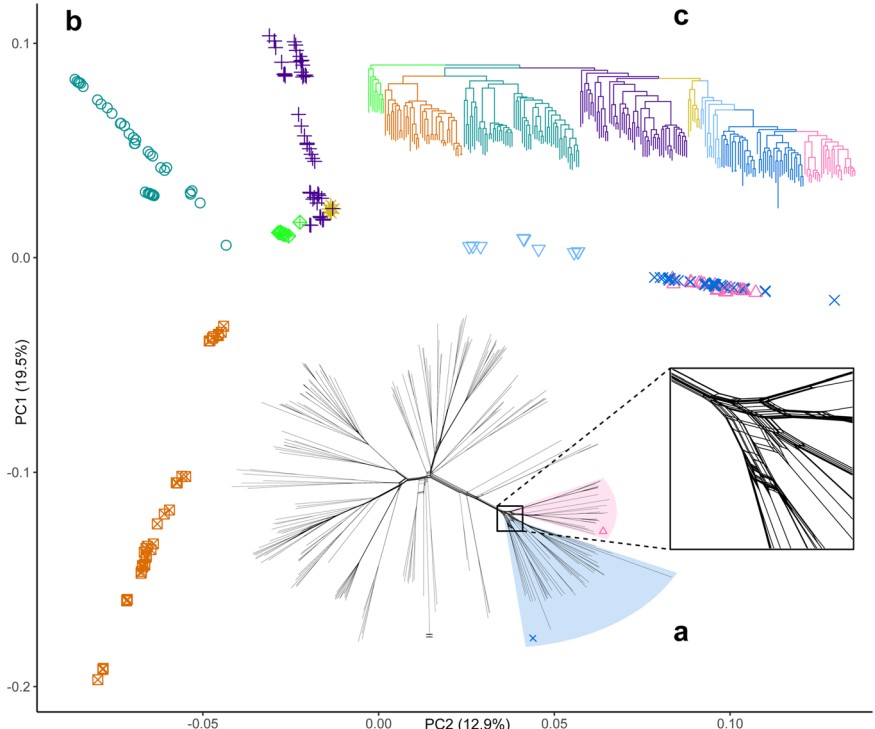

**Fig. 3 | Principal component analysis and phylogenetic reconstructions of single nucleotide polymorphism variation within *Syzygium*. a** A NeighborNet phylogenetic network shows considerable character discordance among genome-wide SNPs that may be indicative of incomplete lineage sorting. This discordance is particularly noteworthy at the highly webbed base of the *Syzygium grande* group (close-up view in square inset; see also the labelled network in Supplementary Fig. 89). **b** PCA of principal components 1 and 2 of *Syzygium* subg. *Syzygium* individuals. Clinal patterns are readily observed; the *Syzygium grande* group (centred around 0.10 on PC2) is comprised of a medium-blue paraphyletic grade subtending a pink terminal lineage, as shown in (**c**), the RAxML SNP tree, which is colour-coded following the PCA. Shading of two groups on (**a**) matches the colour coding on the tree as well as the colours and symbols on the PCA plot. Small matching symbols in shaded areas are shown for clarity. Edge with two horizontal lines at tip represents the outgroup taxon, *Syzygium rugosum*, clipped for length. Source data are provided as a Source Data file.

whole-chromosomal variations as well as repeat-masked data, the latter to ensure that distinct patterns obtained were not solely related to ambiguous mappings due to different transposable element families. The *Syzygium grande* group characteristically appears as a tight cluster across different corners on the 11 chromosomes (Supplementary Figs. 88–100). However, in some of these collections of windows, the group is unresolved from its closest outgroups and from the rest of *Syzygium* subg. *Syzygium*; in other corners, these outgroups are poorly distinguished from the remainder of the subgenus, while the *S. grande* group stays distinct. We infer that these results support the hypothesis of underlying ILS−i.e., regional block-wise genomic distinction vs. indistinction of these taxa, as reflected by the many-paralleled edges of their corresponding stem lineages in the NeighborNet result (Fig. 3a).

**Principal component analysis reveals clinal patterns reflective of isolation by distance**

We further studied the SNP data genome-wide using standard PCA[59,60]. Plots of principal components focussing on *Syzygium* subg. *Syzygium* illustrated clear clines (Fig. 3b) that mostly correspond to sublineages on the BUSCO and SNP trees (Fig. 3c and Supplementary Figs. 101–121). Several filtrations of data (Supplementary Table 2), including analyses of homozygous sites only (as well as checks for coverage that suggested no apparent biases), yielded similar results and therefore increased confidence that the clinal patterns were not artefactual (Supplementary Figs. 107–121). A simple explanation for these linear gradations is that allelic variation in *Syzygium* became fixed in consecutive speciation events, along an ongoing cladogenetic process. The PCA analysis highlights that different lineages within the *S. grande* group partly overlap (Fig. 3b and Supplementary Figs. 122–127), consistent with short internal coalescence branch lengths on the BUSCO

tree. In other words, the clinal patterns may reflect a neutral process akin to isolation by distance[59–62] (IBD; see ref. 63), for example, comprising serial founder events in an island-hopping model of geographic speciation[3] (but see ref. 64). Similar clinal variation among Big Island (Island of Hawai'i) accessions of a closely related Myrtaceae species, *Metrosideros polymorpha* (see Fig. 1C of ref. 10), might also reflect simple IBD processes in its extremely young and rapidly expanding/dissecting volcanic environment.

Allopatric speciation does not necessarily require adaptive differences, only the null model of reproductive isolation and genetic drift[65]. The possibility of entirely neutral phenotypic clines forming in a model of progressive cladogenesis, such as we hypothesise here for diagnosable *Syzygium* species, may attest to IBD, and reflect environmental gradients that accompany spatial population expansion, or even involve admixture between previously isolated populations or clades[63]. However, many *Syzygium* species are sympatrically distributed, which, if the splits observed were time-coincident or nearly so, could suggest that ecological speciation[66–68] (and therefore adaptive differences, such as flowering allochrony and other gene flow barriers) could also be operative. Even weak selection via such local adaptation can significantly speed up an entirely drift-based geographic speciation process[65]. For example, the phylogenetic results presented here show that *Syzygium* species sympatric in the Bukit Timah and Danum Valley forest plots are broadly distributed across the phylogenetic tree, but there are also clear clusters of species from these plots within some subclades (Fig. 2, Supplementary Note 1 and Supplementary Table 2). However, there is reason to suspect that sampling biases from within larger species ranges may influence this clustering, since Bukit Timah- or Danum-enriched clades are not entirely autochthonous to these plots, but also distributed elsewhere.

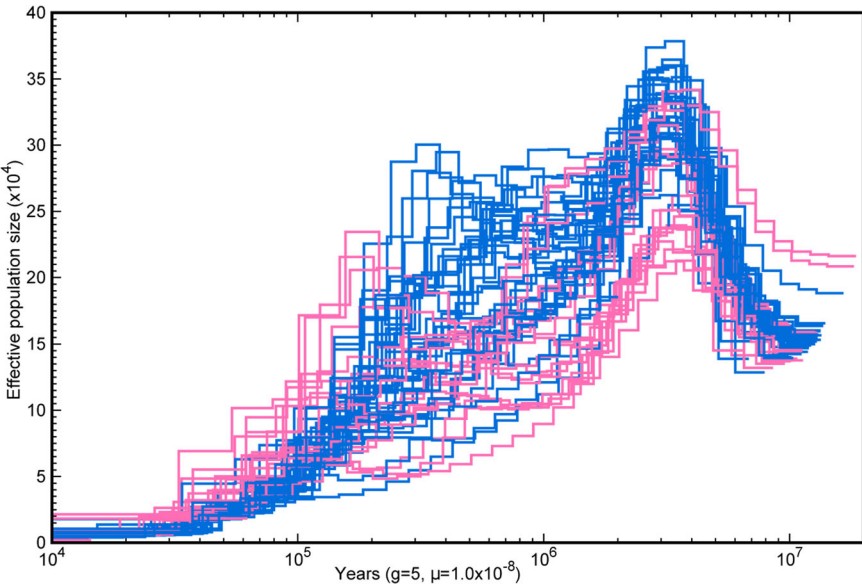

**Fig. 4 | Genomic palaeodemography of *Syzygium* accessions from the *S. grande* group.** Pairwise Sequentially Markovian Coalescent analyses, coloured by groups/clades in Fig. 3. Source data are provided as a Source Data file.

For example, we sampled a *S. barringtonioides* specimen from Brunei that groups within an otherwise Danum cluster (both nonetheless being Bornean), and the *S. chloranthum-S. cerasiforme* clade that was largely sampled from Bukit Timah also contains *S. ampullarium*, which was collected in Borneo. As such, considerable reproductive isolation and lineage diversification likely occurred prior to many migrations into sympatric niches. The clearest inference from Fig. 2 is therefore that the Bukit Timah and Danum *Syzygium* floras are assemblages of phylogenetically- and time-diverse lineages.

### Demographic analysis of the Syzygium *grande* group also implies rapid diversification

Pairwise Sequentially Markovian Coalescent (PSMC) analysis uses the two haploid genomes present in each collection of reads for a given diploid individual to estimate past effective population sizes over time. We ran PSMC demographic curves for most individuals in the closely interrelated *Syzygium grande* group using Illumina reads mapped against each taxon's own MaSuRCA genome assembly. We scaled the time and $N_e$ axes of the demographic reconstructions uniformly by employing an approximate *Syzygium* generation time of 5 years (roughly the median of the data in Supplementary Table 3) and a mutation rate of 1E−08, in line with previous work on woody plants[69]. Comparing demographic curves together, all individuals appear to follow similar trajectories wherein the genetic variation in all taxa coalesces between 9 and 20 million years ago (Mya), followed by a peak in $N_e$ at 3–4 Mya, various $N_e$ fluctuations/crashes in intermediate times from 1 to 0.1 Mya, followed by strong $N_e$ collapse in recent-most time (Fig. 4). The impression that the demographies seem largely alignable in gross aspect while differing in ancient coalescence times, may reflect their joint membership in a stem lineage as well as real generation time differences among the taxa, or differences in past heterozygosity levels[70]. Indeed, maximum time at coalescence strongly correlates with overall heterozygosities of individuals (from SNP calls) as well as numbers of segregating sites (from reads mapping to individual genomes only; Supplementary Fig. 128).

Moreover, PSMC curves for many taxa in the pink terminal lineage in Fig. 3c converge together at lower $N_e$ in ancient to intermediate times than do most individuals in the blue paraphyletic group that subtends it, which mostly follow higher $N_e$ trajectories. These distinctions visible as early as 10 Mya suggest that the pink lineage may

have begun splitting from within the blue group as early as then, while the $N_e$ fluctuations closer to the present may represent the period of rapid cladogenesis reflected in the "fan-like" reticulate base of the *S. grande* group that is visible in the NeighborNet result. Phylogenetic resolution of rapid splits such as these can be particularly confounded by ILS, which by coalescent theory may itself be exacerbated by any $N_e$ size increases. The final $N_e$ crashes closest to the present may in turn mark the individuation of the lineages (e.g., via founder effect[71]) visible past the stage of the NeighborNet fan (i.e., the tips extending from its basal web).

### *Syzygium* radiated multiple times from Sahul into Sunda and elsewhere

In a review on the origins and assembly of Malesian rainforests[72], *Syzygium* was highlighted as a key genus for understanding the floristic evolution of the region. Formal biogeographic analyses using the BioGeoBEARS[73] and RASP (Reconstruct Ancestral State in Phylogenies)[74] software each demonstrate, despite limited taxon sampling of outgroups, that the genus *Syzygium* is of Sahul origin, i.e., centred on Australia and New Guinea (Supplementary Figs. 129 and 130; source data are provided at Dryad, https://doi.org/10.5061/dryad.h18931zpw). This finding is consistent with previous work on *Syzygium* and Myrtaceae as a whole, which similarly finds Sahul as the ancestral area[75]. We also generated a dated ultrametric SNP tree to provide split and crown group times for subclades and species diversifications (Supplementary Fig. 131; source data are provided at Dryad, https://doi.org/10.5061/dryad.h18931zpw). We used as a calibration point the minimum and maximum ages of a fossil assignable to *Syzygium* subg. *Acmena* (20.9–22.1 Mya)[76]. The crown group of the entire genus *Syzygium* is dated at 51.2 Mya, and the crown groups of subgenera *Sequestratum*, *Perikion*, *Acmena*, the *S. rugosum* clade, and *Syzygium* date to 34.2, 24.1, 15.8, 7.0 and 9.4 Mya, respectively (Supplementary Fig. 131). As such, *Syzygium* itself dates to before the Sunda-Sahul convergence which occurred ~25 Mya[77], with most subgenera diversifying after the convergence.

Repeated invasions both westward and northward from Sahul that correspond with species diversifications are clearly apparent. For example, parallel migrations into Sunda occurred at least 12 times (Fig. 2), sometimes corresponding with large radiations, but only within *Syzygium* subg. *Syzygium* (Supplementary Figs. 129 and 130).

The earliest migration to Sunda was by 17.1 Mya, the crown group age for the Sunda half of the first split in *Syzygium* subg. *Sequestratum* (Supplementary Fig. 131). The *S. rugosum* clade migrated to Sunda by 7.0 Mya, and *Syzygium* subg. *Perikion* had entered Sunda (Peninsular Malaysia) and later migrated to Sri Lanka by 3.0 Mya. Within *Syzygium* subg. *Acmena*, Sunda had been accessed by 390 Kya. *Syzygium* subg. *Syzygium* is resolved as having a Sahul origin, with a crown group age of 9.4 Mya (corresponding to the young end of PSMC curve coalescences for the *S. grande* group; see above and Fig. 4). Following Hall's[78] land/sea level reconstruction at 10 Mya, entry of subgenus *Syzygium* into Sunda, potentially via the Sula Spur, may have involved considerable island hopping from Sahul. As many as seven invasions of Sunda occurred, at least three of which (according to our sampling) resulted in hyperdiverse subclades. The earliest Sunda migrations within the type subgenus involved the hyperdiverse *Syzygium pustulatum* group, with a minimum crown group age of 2.8 Mya, and the large *S. creaghii* group, which has a similar minimum crown age of 2.5 Mya (Supplementary Fig. 131). These lineages entered Sunda following the New Guinea uplift, which began about 5 Mya[79], possibly correlating with population expansions seen around this time in PSMC curves for the extremely diverse *Syzygium grande* group. The *S. grande* lineage migrated much later from Sahul into Sunda by 165 Kya, overlapping with the PSMC $N_e$ fluctuations seen in intermediate times (Fig. 4). It subsequently radiated broadly and very recently into the North Pacific (by 14.6 Kya), the Indian subcontinent (by 21.9 Kya), and from there on to Africa (by 6.5 Kya). These recent dates correspond well with the individuation of clades within the *S. grande* group inferred from NeighborNet (Fig. 3a), and discussed above in reference to population crashes in PSMC analyses (Fig. 4). The *Syzygium pustulatum* and *S. creaghii* groups, which are also marked by fan-like radiations in the NeighborNet analysis (see labelled network in Supplementary Fig. 89), unlike the *S. grande* group, do not show considerable character incongruence suggestive of ILS at its base. The *Syzygium pustulatum* group and smaller and late-migrating *S. jambos* group (the latter having entered Sunda by 123 kya; Supplementary Fig. 131) also represent rapid diversifications into Sunda with significant tree-like structure at their stem-lineage bases in the NeighborNet analysis. The last 1 Mya in Southeast Asian biogeography was marked by cyclical sea level changes that repeatedly divided and rejoined vegetation[80,81], and the minimum invasion dates for the *Syzygium grande* and *S. jambos* groups correspond with periods when sea levels were lower than today[82] and therefore lowland rainforest vegetation more continuous. To summarise, parallel dispersals from Sahul into Sunda and beyond sometimes correlate with what appear to be rapid radiations that at least in one case, the *Syzygium grande* group, appears to have been marked by significant ILS.

## Morphological transitions may accompany some *Syzygium* species diversifications

We thereafter sought, using Mesquite[83] parsimony optimisations, to provide qualitative first approximations of morphological trait evolution and accompanying ecological variables that might correspond with these East-to-West migrations (source data are provided at Dryad, https://doi.org/10.5061/dryad.h18931zpw). We employed the BUSCO species tree for this exercise to minimise any topological biases that might arise from ILS. An interesting trait is the presence of a pseudocalyptrate (or "calyptrate" in *Syzygium barringtonioides* and *S. perspicuinervium*) versus free corolla (Fig. 5a–c, Supplementary Note 2 and Supplementary Figs. 132 and 133). A pseudocalyptrate corolla, which is relatively common among genera of Myrtaceae, describes a perianth that is variously fused into a cap-like structure that may protect developing stamens from predation, degradation by desiccation, or fungal rot[84]. As determined previously, based on PCR marker phylogenies[15,16] and ontogenetic studies[84], pseudocalyptrate corollas evolved convergently in several *Syzygium* groups. One remarkable

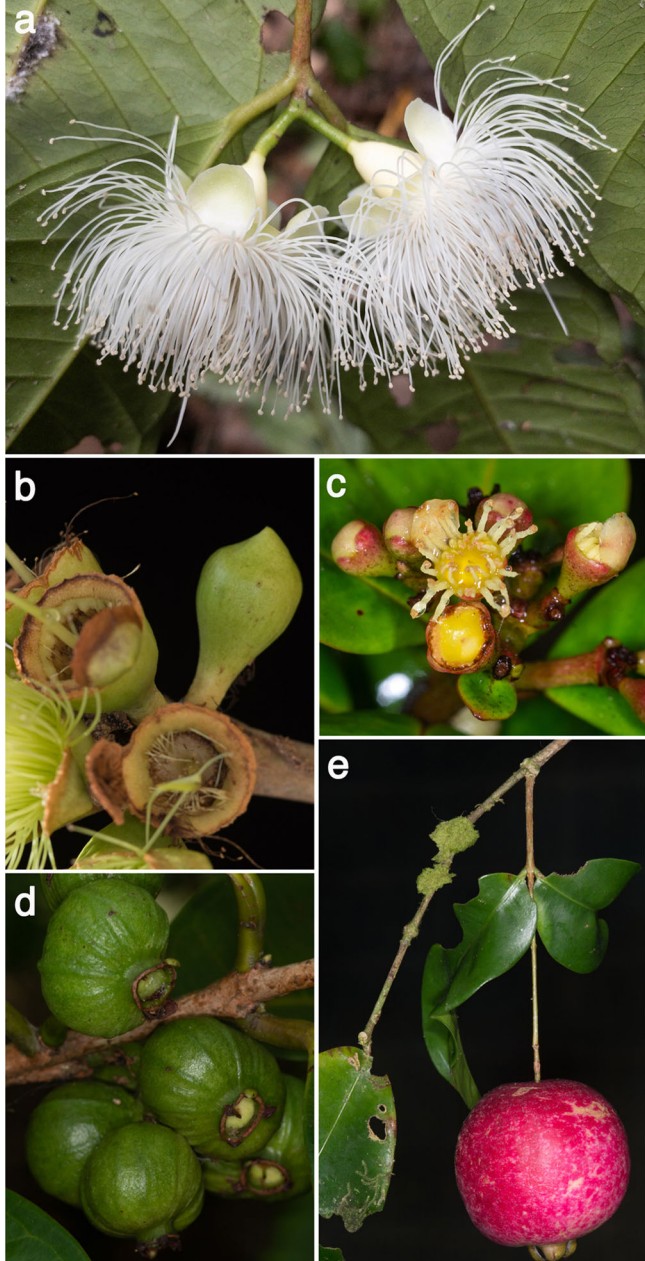

**Fig. 5 | Reproductive trait diversity in the genus *Syzygium*,** as examined to reconstruct ancestral states using Mesquite. **a** Free petals (*Syzygium pendens*); **b** Calyptrate calyx (*Syzygium paradoxum*); **c** Pseudocalyptrate corolla (*Syzygium adelphicum*); **d** Fruits maturing green (*Syzygium* cf. *dyerianum*); **e** Pendulous inflorescence or infructescence (*Syzygium boonjee*). Photograph credits: YWL (**a**)–(**e**).

transition from free to pseudocalyptrate corollas appears at the base of the *Syzygium grande* group; indeed, it was apparently fixed first in its outgroup taxa (Supplementary Fig. 133). Several evolutionary reversals thereafter to free corolla lobes occurred, including one reversal that marks a large sublineage of the *Syzygium grande* group including 75 species as well as *S. grande* itself. The *Syzygium creaghii* and *S. jambos* groups have free corollas, but the *S. pustulatum* group may have been primitively pseudocalyptrate. Regardless, this trait seems highly labile within *Syzygium*, and other than the possible exception of the *S. grande* group's ancestral state, there is no clear connection with Sahul-to-Sunda diversification. Interestingly, the most-parsimonious resolution of green fruits as ancestral to this clade and some of its outgroup species accompanies diversification of the *Syzygium grande* group into Sunda (Fig. 5d and Supplementary Fig. 134). Later most-

parsimonious state transitions from green to purplish-black fruit are also noteworthy in the group. We speculate that this combination of traits—pre-anthesis protection by pseudocalyptrae and bearing of green to purplish-black fruits that attract far-flying birds or bats[85–87]—may have together pre-adapted this group to broad migration.

One other trait of note that marks large diversifications is the presence of pendulous inflorescences, which characterises the *Syzygium creaghii* group and largely marks the *S. longipes* group (Fig. 5e, Supplementary Note 2 and Supplementary Fig. 135). This trait is correlated with large fruits, often fleshy, which are known to reflect a specialised fruit display and dispersal strategy called flagellichory that increases fruit display for echolocating bats[88], other flying/arboreal vertebrates[18] or large vertebrate browsers (e.g., cassowaries[89]).

## Implications from *Syzygium* for understanding species radiations

Here, we have explored species diversification patterns and their drivers in the world's most species-rich tree genus, *Syzygium*. We generated a high-quality reference genome for *Syzygium grande*, the sea apple, and shotgun sequenced more than 15% of the species of this large genus to study their phylogenomic relationships. Through this extensive sampling of *Syzygium* diversity, we were able to solidify major clade relationships within the genus, currently recognised as subgenera, and, within *Syzygium* subg. *Syzygium*, provided unprecedented clarity on subclades that may become sectional units in the future. We discovered that many *Syzygium* species, particularly within *Syzygium* subg. *Syzygium*, likely branched from one another in rapid succession, yielding radiations of morphological and ecological diversity. One example was a group of species containing *Syzygium grande* itself that was marked by extremely short coalescence time intervals in our BUSCO species tree; this result was matched by highly networked edges at the base of the group in our NeighborNet analysis, which reflects underlying incongruence in the data. Since none of the $f_3$ tests showed admixture, we interpret such webbed stem lineages in the NeighborNet network to reflect incomplete lineage sorting during rapid species radiation. PCA analysis of our samples illustrated clines of fixed allelic variation arrayed by *Syzygium* sublineage, possibly reflecting that a simple process of neutral geographic speciation predominated during most of the group's cladogenesis. Plotting occurrences of species native to Singapore's Bukit Timah Nature Reserve and East Malaysia's Danum Valley Conservation Area illustrated that large-scale lineage diversification occurred before sympatric occupation of these habitats to generate diverse, closely associated *Syzygium* floras. As such, the immense radiation of the world's largest tree genus may serve as a model for further detailed research, for example at the population level—integrating transcriptomic, proteomic, and metabolomic data—to explore actual mechanisms underlying morphological and ecological specialisation during a diversification that rivals any others under current study.

## Methods

### Oxford Nanopore sequencing of *Syzygium grande*

Young leaf tissue and twigs of *Syzygium grande* from a cultivated individual (Gleneagles Hospital, along Napier Road, Singapore; *Low s.n.* [SING]) were gathered, cleaned and flash frozen in liquid nitrogen, and then stored in −80 °C prior to extraction. About 10 g of flash frozen tissue was used for high-molecular-weight (HMW) genomic DNA isolation. The first step followed the BioNano NIBuffer nuclei isolation protocol in which frozen leaf tissue was homogenised in liquid nitrogen, followed by a nuclei lysis step using IBTB buffer with spermine and spermidine added and filtered just before use. IBTB buffer consists of Isolation Buffer (IB; 15 mM Tris, 10 mM EDTA, 130 mM KCl, 20 mM NaCl, 8%(m/V) PVP-10, pH 9.4) with 0.1% Triton X-100, and 7.5% (V/V) β-Mercaptoethanol (BME) mixed in and chilled on ice. The mixture of homogenised leaf tissue and IBTB buffer was strained to remove undissolved plant tissue. 1% Triton X-100 was added to lyse the nuclei before centrifugation at 2000×*g* for 10 min to pellet the nuclei. Once the nuclei pellet was obtained, we proceeded with cetyl-trimethylammonium bromide (CTAB) DNA extraction with modifications for Oxford Nanopore sequencing[90]. The quality and concentration of HMW genomic DNA was checked using a Thermo Scientific™ NanoDrop™ Spectrophotometer, as well as on agarose gel electrophoresis following standard protocols. Genomic DNA obtained was further purified with a Qiagen® Genomic-Tip 500/G following the protocol provided by the developer.

The purified genomic DNA sample obtained was sequenced on the Oxford Nanopore Technologies (ONT) PromethION platform. We generated 60,136,770,518 bp of Nanopore reads with a read length N50 of 9382 bp and an average read quality score of 6.5. Raw ONT reads (fastq) of *Syzygium grande* were filtered prior to assembly using seqtk[91] such that only reads 35 kb or longer were used for genome assembly, which was performed using wtdbg2[30] version 2.2 with flags -p19 -AS2 -e2. The genome consensus was also generated with wtdbg2. Consensus correction was performed with the input ONT reads and three rounds of racon[92]. The assembly generated was polished with Pilon[93] using 30 Gb of 2 × 150 paired Illumina HiSeqX reads of *Syzygium grande* that were trimmed and filtered. The assembly of *Syzygium grande* comprised 1669 contigs with an N50 length of 556,915 bp. The assembly was filtered for organellar and contaminating contigs using the blobtools[94] pipeline, resulting in the removal of 30 out of 1669 contigs. Next, purge haplotigs[95] was used to identify 744 contigs contributing to a diploid peak, which were then removed. These contigs comprised <40 Mb of the genome assembly. This filtered primary assembly was thereafter scaffolded into chromosomes by Dovetail HiC technology[31]. The final scaffolded assembly size was 405,179,882 bp.

### Transcriptome assembly and annotation of the *Syzygium grande* genome

Transcriptome assembly was carried out for 3 RNASeq libraries (S1: young leaves, S2: mature leaves, S3: twig tips; sequencing performed by NovogeneAIT) separately using an in-house custom assembly pipeline. The first step involved de novo assembly for multiple kmer values— 51, 61, 71, 81, 91, 101 using TransAbyss[96] v2.0.1, and for kmer value 25 using Trinity[97] v2.8.5. The second step comprised genome-guided assembly using StringTie[98] v2.0. The input for this second step involved aligning the RNASeq reads against the reference genome using HISAT2[99] v2.1.0. The third step encompassed combining all the results from the first and the second steps using EvidentialGene[100] v2018.06.18 to obtain a final high-confidence transcriptome assembly. S1 produced 57,746 transcripts (BUSCO completeness 92.9%), S2 produced 56,536 transcripts (BUSCO completeness 94.6%) and S2 produced 64,163 transcripts (BUSCO completeness 94.1%).

The genome annotation of the reference *Syzygium grande* genome was carried out using an in-house custom annotation pipeline. The first step involved the preparation of a de novo repeat library using RepeatModeler v1.0.11. This library was used to mask the repetitive regions in the genome assembly using RepeatMasker[101] v4.0.9 resulting in 45.09% of the genome being masked. The second step was the gene prediction step, based on a modular approach using three different gene predictors gene markers, braker (using the three RNASeq libraries) and GeMoMa[102] (using gene models from the model species *Arabidopsis thaliana* [TAIR10] and *Populus trichocarpa* [v3.1]). Additionally, the spliced transcript aligner PASA[103] (using transcripts from the three RNASeq libraries) was used to generate evidence for gene structures. These results were then combined using the combiner tool EvidenceModeler[104] to produce a single high confidence final prediction of 39,903 gene models with a BUSCO completeness score of 86.6%. A graphic workflow of these procedures is presented in Supplementary Fig. 136. Please see additional details in Supplementary Note 3.

## Genome structural analyses

The chromosome-level *Syzygium grande* genome assembly and annotation were uploaded to the online CoGe comparative genomics platform (https://genomevolution.org/coge/GenomeInfo.pl?gid=60239)[105]. Syntenic dot plots and data for synonymous substitution rate (Ks) calculations were derived from CoGe SynMap[105] calculations using default settings, with CodeML set to "Calculate syntenic CDS pairs and colour dots: Synonymous (Ks) substitution rates". Ks data were collected from corresponding downloads at the "Results with synonymous/non-synonymous rate values" tabs. Each pairwise SynMap analysis (including self:self) was performed for the following species and CoGe genome IDs: *Syzygium grande* (id60239), *Eucalyptus grandis* (id28624), *Punica granatum* (id61248), *Populus trichocarpa* (id25127), *Vitis vinifera* (id19990). Syntenic dot plots from SynMap were further investigated for synteny relationships within and between species using the FractBias tool[106]. FractBias mappings for fractionation profiles between species were generated using Quota Align syntenic depth of 2:1 for *Syzygium*, *Eucalyptus,* and *Punica* against *Vitis* (analyses can be regenerated at https://genomevolution.org/r/1ig9p, https://genomevolution.org/r/1ig9r, and https://genomevolution.org/r/1ig9o, respectively), max query chromosomes = 100, max target chromosomes = 25, and "Use all genes in target genome". For *Populus* against *Syzygium* (which can be regenerated at https://genomevolution.org/r/1ig9q), mapping of the former assembly against the latter used a Quota Align syntenic depth of 2:2 and the same options as described above for depth 2:1. Density plots (both histogram and smoothed curve) of Ks values for syntenic paralogs were generated in R[107] using the tidyverse[108], ggplot2[109], RColorBrewer[110], ggridges[111], and ggpmisc[112] packages. Ks peaks were calibrated by their shared *gamma* hexaploidy event using the method described by Wang et al. [43].

## Illumina sequencing of *Syzygium* and outgroup individuals

A total of 289 *Syzygium* individuals were selected to represent the six subgenera recognised by Craven and Biffin (2010)[17], across its natural distribution from Africa to the Indian subcontinent, through the Indomalaya region and into the Pacific. Three outgroup taxa in Myrtaceae, *Metrosideros excelsa*, *M. nervulosa* (tribe Metrosidereae) and *Eugenia reinwardtiana* (tribe Myrteae), were also sampled. Most of the 292 samples used in this study were freshly collected in the field, utilising the silica gel teabag method for preserving plant DNA[113], between 2017 and 2019 either from collecting expeditions conducted in Singapore, Australia, Brunei, Indonesia (West Papua and Papua provinces) and Malaysia or from cultivated specimens in the Singapore Botanic Gardens (Singapore), Bogor Botanical Garden (Bogor, Indonesia), Cairns Botanic Gardens (Queensland, Australia) and Royal Botanic Gardens, Kew (UK).

Approximately 20 mg of silica-dried leaf tissue were sampled for genomic sequencing. Plant tissue was ground to a fine powder using Omni International Bed Rupture Homogeniser. DNA isolation was carried out at the molecular lab of the Singapore Botanic Gardens using the Qiagen DNeasy® Plant Mini Kit, following the protocol provided by the manufacturer. In rare cases, DNA yields were low when obtained from Qiagen DNeasy® Plant Mini Kit; hence for these problematic samples, the Qiagen DNeasy® Plant Maxi Kit was used instead. Quality and concentration of DNA aliquots were checked using a Thermo Scientific™ NanoDrop™ Spectrophotometer before submission to NovogeneAIT (Singapore) for QC, library construction and sequencing of 30 Gb each (150 × 150 paired ends) on an Illumina HiSeqX.

## Assembly, BUSCO QC, and species tree phylogeny of the resequenced *Syzygium* and outgroup accessions

The 292 Illumina resequenced accessions were assembled using MaSuRCA[33] v3.3.1 with library insert average length of 350 bp and a standard deviation of 100 bp. The genome completeness percentages were estimated using BUSCO v4.0.2 based on eudicots_odb10 database.

The phylogeny for the *Syzygium* and outgroup species was estimated using the BUSCO genes. Two species tree versions were estimated. The first tree was estimated using 229 BUSCO genes that were complete and found in all 292 species. The second tree was estimated using 1227 BUSCO genes that were present in 286 species and above.

The species tree generation was constructed using an in-house phylogeny pipeline. The first step involved extraction of BUSCO genes from all resequenced individuals, generating a multi-fasta file for each BUSCO gene containing a representation of that gene from the available species. The second step involved performing multiple sequence alignment (MSA) for each BUSCO multi-fasta file using MAFFT[114] v7.407. The resulting MSA files were used to generate gene trees using RAxML[47] v8.2.12. These gene trees were concatenated and sent as a single input to ASTRAL[48] v5.15.1 to generate the final species tree.

## Mapping the resequenced individuals to the *Syzygium grande* reference genome

The 30 Gb each of raw Illumina reads was trimmed to remove adapters using default settings of Trimmomatic[115] version 0.38. Following trimming, the samples were mapped using bwa mem[116] (version 0.7.17), and the subsequent bam files were filtered for a quality score of 20 using samtools[117] view and sorted using samtools sort. Picard MarkDuplicates (version 2.7.1; https://broadinstitute.github.io/picard/) was used to remove PCR duplicates from the mapped reads. Depth and width of mapping coverage were calculated using BEDTools[118] version 2.23.0.

## SNP calling and statistics

SNP calling was performed using GATK version 3.8 in ERC mode for each sample. GenotypeGVCFs were used to call joint genotypes; due to RAM and time limitations, this was split into 70 intervals using the −L flag. To combine the 70 files, GatherVcfs was used to generate a VCF file. As a quality control, GATK VariantFiltration was used with the following filter expression based on GATK recommendations: 'QD < 2.0 || FS > 60.0 || MQ < 50.0 || MQRankSum < −12.5 || ReadPosRankSum < −8.0 || SOR > 4.0'. Further filtrations were carried out in VCFtools[119] (version 0.1.13) to create various datasets for downstream analyses (Supplementary Table 2). The −plink flag was applied to generate .ped and .map files, and also −recode was used to generate a filtered vcf file. SNP statistics were calculated through vcftools options −het and −singletons for dataset FRSA-1. The output was subsequently plotted using the R package ggplot (https://github.com/tidyverse/ggplot2).

## RAxML SNP tree

A pseudo-alignment of SNPs was generated for phylogenetic reconstruction for dataset FRSA-1. The plink.ped file was used to convert into fasta input for RAxML. Only variable SNPs were retained for a total of 2,384,277 SNPs. A maximum likelihood tree was generated using RAxML version 8 including adjustments for ascertainment bias (−asc-corr lewis) and 500 bootstraps. Trees were viewed and edited using FigTree[120].

## Plastome assembly and phylogenetic tree

We filtered and removed nuclear reads from the *Syzygium grande* Nanopore assembly and constructed a complete chloroplast genome of 158,980 bp in length. This genome was used as a reference to examine phylogenetic relationships of *Syzygium* based on the plastome. Before mapping Illumina reads of 289 *Syzygium* individuals and three outgroup taxa (two *Metrosideros* and one *Eugenia*) to the reference, one of the Inverted Repeat regions (IRs) was removed to prevent sequence calling bias. The combined DNA alignment file of the 292 individuals was then subjected to an ML analysis using RAxML with 1000 bootstrap replicates.

### NeighborNet analysis

We used the NeighborNet[50,121] approach to assess incongruence within our SNP data set for *Syzygium* subg. *Syzygium* and *S. rugosum* as an outgroup taxon (FRSA-5). We used SplitsTree[122] version 4.17.0 to calculate the network with LogDet[123] distances.

### ADMIXTURE analysis

We ran ADMIXTURE[53] (version 1.3) for *K* values 5–15 and used the –cv option to find the best (lowest) *K* value for the number of ancestral populations (Supplementary Figs. 10 and 11). The results were then plotted using the barplot function in R.

### Local PCA

Single nucleotide polymorphisms (SNPs) called against the draft assembly were transferred to the Hi-C scaffolded assembly through the use of Minimap2[124], transanno (https://github.com/informationsea/transanno), and LiftoverVcf[125]. The BED file needed to remove SNPs from repeat regions was generated using convert2bed[126]. SNPs from repeat regions were removed using the VCFtools–exclude-bed option[119]. The VCF file was divided by the 11 pseudomolecules using HTSlib[127], and converted to BCF format and indexed using BCFtools[128]. Local PCA was carried out using the R[107] lostruct package[58] and window size was chosen to include about 1000 SNPs per window as recommended by the authors.

### PCA

The eigensoft[129] package (version 6.1.3) was used to convert plink.map and .ped files into .ind, .geno and .snp files. Thereafter, the smartpca.perl script was used to run PCA for PC1 to PC10 under default parameters for datasets FRSA-1 (all *Syzygium*) and FRSA-3 (*Syzygium* subg. *Syzygium*). Taxa that were removed using the smartpca default five rounds of outlier removal shown in red (Supplementary Fig. 101). In addition, three separate PCA checks were performed to confirm that clinal results were not artefactual. PCA was run with SNPs using (i) a more stringent minimum depth of coverage of 20 (Supplementary Figs. 112–116), (ii) homozygous sites only (Supplementary Figs. 117–121), and (iii) LD correction turned on using the nsnpldregress option in the smartpca programme to control for linkage (Supplementary Figs. 107–111). In order to search for possible correlations, the PCA was coloured in numerous ways, by geography, by ecoplots, and also according to ADMIXTURE *K* = 14 ancestral groups (Supplementary Fig. 11), using the scatterpie package in R.

### $f_3$ statistics

Dataset FRSA-5 was used to formally test for admixture using the $f_3$[56] statistic implemented in the qp3pop (version 650) function of the AdmixTools package (https://github.com/DReichLab/AdmixTools). A total of 7,195,530 SNPs were used to test 3,889,44 triplets, every possible combination within the *Syzygium grande* group. We then applied a FDR correction to the *Z*-scores using a custom R function developed as part of the silver birch genome project[130]. Next, heatmaps were plotted in R for each target.

### PSMC

We used PSMC[131] to infer past demographies for most members of the *Syzygium grande* group. To accomplish this, we mapped trimmed reads for each sample to its de novo MaSuRCA[33] assembled genome, using the same mapping parameters used for *S. grande* reference mapping. Consensus sequences were called using samtools[117] mpileup to generate diploid sequences for input to PSMC, with all parameters set to default. Demographic curves were subsequently plotted using the psmc_plot.pl script. As a first quality control, we only included samples with de novo assemblies that had N50 > 10,000 basepairs and a BUSCO completeness score >80%, which excluded additional eight samples. Following plotting, for clarity, we removed an additional six samples that deviated strongly from the general trends of the clade. The values of sum_n, the number of segregating sites, were extracted from the .psmc files for each sample. The maximum coalescence date and its corresponding $N_e$ values were extracted from the plot files. Since strict filtering of SNPs lowers heterozygosity drastically, the dataset FRSA-GATK was used to calculate heterozygosity using the –het option in VCFtools. To plot the correlation matrix, the pairs.panels function from the R package psych was used. The parameter lm was set to true to display linear regression fit, and ci was set to true to display confidence intervals. To display *R*-squared values, the source code was edited. Stars were assigned to the following categories based on *p*-value significance (<0.001***, <0.01**, and <0.05*).

### Biogeography

An ultrametric dated SNP phylogeny was generated both for phylogenetic dating and biogeographic reconstruction. The SNP tree was employed since ASTRAL species tree branch lengths are not interpretable for ultrametric conversion (see the comment from the ASTRAL developer on GitHub: https://github.com/smirarab/ASTRAL/issues/37). The function chronos() in the R package ape[132] (version 3.5.2) was used to create the ultrametric tree. The model used was correlated, and the calibration applied a minimum of 20,900,000 and maximum of 22,100,000 years at the node for which *Syzygium* subg. *Acmena* and *S*. subg. *Syzygium* share a common ancestor. This calibration is based on a *Syzygium* fossil (*S. christophelii*) found in New South Wales, Australia[76]. The ultrametric tree was also used as input to search for and execute the best model using RASP[74] and BioGeoBEARS[73] (which was BAYAREALIKE + J in both cases). Each of the 292 samples was assigned to one or more of eight geographic regions (Africa, India, Mainland Asia, Sunda, Sahul, Wallacea, Zealandia, and Pacific islands) based on their distribution patterns, and RASP and BioGeoBEARS were both restricted to a maximum of six areas at each node.

### Character evolution with Mesquite

States for three morphological characters—specifically (i) inflorescence habit (erect vs. pendent), (ii) shedding fused corolla present as a true calyptra, a pseudocalyptra, vs. corolla free at anthesis, and (iii) mature fruit colour (green, white or cream, black, pink, purple, red, brown, orange, yellow, blue, or grey)—were gathered from living material, herbarium specimens, published flora accounts, and species protologues. The categorical morphological characters were coded into the form of numbers 0–9 and/or letters a–z. The ASTRAL tree of BUSCO genes, and selected traits, were loaded into Mesquite[83] version 3.61 and the Trace Character Evolution option with parsimony was selected to predict ancestral states.

### Reporting summary

Further information on research design is available in the Nature Research Reporting Summary linked to this article.

### Data availability

The genome data generated in this study has been deposited in the NCBI database under accession code PRJNA803434 and BioSample ID SAMN29207412. The *Syzygium grande* genome assembly and annotation are also available on CoGe [https://genomevolution.org/coge/GenomeInfo.pl?gid=60239]. Processed data generated in this study and used for main text figures are provided in source data files. Additional processed data are available at Dryad [https://doi.org/10.5061/dryad.h18931zpw]. Source data are provided with this paper.

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

## Acknowledgements

Y.W.L. was supported by a postgraduate scholarship research grant from the Ministry of National Development, Singapore awarded through the National Parks Board, Singapore (NParks; NParks' Garden City Fund). Principal research funding from NParks and the School of Biological Sciences (SBS), Nanyang Technological University (NTU), Singapore, is acknowledged. We thank Peter Preiser, Associate Vice President for Biomedical and Life Sciences, for facilitating NTU support, and Kenneth Er, CEO of NParks, for facilitating research funding through that organisation. V.A.A. and C.L. were funded by SBS, NTU for a one-year research leave. V.A.A. and C.L. also acknowledge support from the United States National Science Foundation (grants 2030871 and 1854550, respectively). S.R. was supported by a postdoctoral research fellowship under the NTU Strategic Plant Programme. S.R. and N.R.W.C. acknowledge funding from NTU start-up and the Academy of Finland (decisions 318288, 319947) grants to J.S. Fieldwork conducted by Y.W.L. was supported by an Indonesian Government RISTEK research permit (Application ID: 1517217008) and an Access License from the Sabah State government [JKM/MBS.1000-2/2JLD.7(84)]. T.N.C.V. is grateful to the Assemblée de la Province Nord and Assemblée de la Province Sud (New Caledonia) for facilitating relevant collection permits. A.N. was partly supported by the Research Project Promotion Grant (Strategic Research Grant No. 17SP01302) from the University of the Ryukyus, and partly by the Environment Research and Technology Development Fund (JPMEERF20204003) from the Environmental Restoration and Conservation Agency of Japan. Fieldwork in Fiji conducted by R.B. was hosted and facilitated by Elina Nabubuniyaka-Young (The Pacific Community's Centre for Pacific Crops and Trees, Fiji). We thank the NTU-Smithsonian Partnership for tree data obtained for the Bukit Timah Nature Reserve (BTNR) long-term forest dynamics plots. Administrative support provided by Mui Hwang Khoo-Woon and Peter Ang at the molecular laboratory of the Singapore Botanic Gardens (SBG) is acknowledged. Rosie Woods and Imalka Kahandawala (DNA and Tissue Bank, Royal Botanic Gardens, Kew) facilitated additional DNA samples. Daniel Thomas (SBG) and Yan Yu (Sichuan University) commented on biogeographical analyses. NovogeneAIT in Singapore is acknowledged for personalised sequencing service.

## Author contributions

Y.W.L., V.A.A., C.L., E.J.L., D.F.R.P.B., and D.J.M. conceived the study. T.P.M. generated the Oxford Nanopore Technology sequence for and assembled the reference genome. Y.W.L., S.R., N.R.W.C., S.J.F., C.M.T.,

C.L., and V.A.A. analysed the genomic data. Y.W.L. assembled the morphological data with contributions from HDJ on Sri Lankan taxa; Y.W.L., C.M.T., and V.A.A. performed the character evolutionary analyses. Y.W.L. and E.J.L. assembled the biogeographic data; Y.W.L., C.M.T., and V.A.A. carried out the biogeographic analyses. Y.W.L., S.R., C.M.T., C.L., and V.A.A. wrote the first draft of the paper. E.J.L., D.F.R.P.B., J.S., and D.J.M. provided comments on the first draft. Y.W.L., J.A.A., W.H.A., K.A., P.A., A.B., R.E.B., M.C., L.M.C., I.D.C., D.C., A.J.F., P.I.F., D.G., D.J.G., B.G., C.D.H., A.I., B.I., H.D.J., M.A.K., H.S.K., E.K., S.L.K., J.T.K.L., S.M.L.L., P.K.F.L., W.H.L., S.K.Y.L., R.M., W.J.F.M., F.M., W.A.M., A.N., K.M.N., M.N., S.R., R.R., H.R., V.I.S., R.S.S., S.S., L.A.T., A.T.B., T.N.C.V., J.F.W., P.W., D.S.A.W., S.W., J.W.Y., K.T.Y., G.S.W.K., D.F.R.P.B., C.L., E.J.L., and V.A.A. participated in fieldwork and/or provided plant tissue samples or other materials for this work. All authors approved the final paper version.

## Competing interests

The authors declare no competing interests.

## Additional information

[1]Singapore Botanic Gardens, National Parks Board, Singapore, Singapore. [2]Royal Botanic Gardens, Kew, London, UK. [3]School of Biological Sciences, University of Aberdeen, Aberdeen, UK. [4]School of Biological Sciences, Nanyang Technological University, Singapore, Singapore. [5]Organismal and Evolutionary Biology Research Programme, Faculty of Biological and Environmental Sciences, University of Helsinki, Helsinki, Finland. [6]Department of Biological Sciences, University at Buffalo, New York, USA. [7]Brunei National Herbarium, Forestry Department, Ministry of Primary Resources and Tourism, Bandar Seri Begawan, Brunei Darussalam. [8]Bogor Botanical Garden, Bogor, Indonesia. [9]New York Botanical Garden, Bronx, NY, USA. [10]Faculty of Tropical Forestry, Universiti Malaysia Sabah, Kota Kinabalu, Sabah, Malaysia. [11]Northern Territory Herbarium, Department of Environment, Parks and Water Security, Darwin, NT, Australia. [12]Australian Tropical Herbarium, James Cook University, Cairns, QLD, Australia. [13]CSIRO, Land and Water, Tropical Forest Research Centre, Atherton, QLD, Australia. [14]Queensland Herbarium, Department of Environment and Science, Brisbane Botanic Gardens, Brisbane, QLD, Australia. [15]Herbarium Bogoriense, Cibinong, Indonesia. [16]BALITBANGDA Papua Barat, Manokwari, Papua Barat, Indonesia. [17]Universitas Papua, Manokwari, Papua Barat, Indonesia. [18]Department of Plant Sciences, Faculty of Science, University of Colombo, Colombo, Sri Lanka. [19]National Institute of Fundamental Studies, Kandy, Sri Lanka. [20]Pulau Ubin, Conservation, National Parks Board, Singapore, Singapore. [21]Asian School of the Environment, Nanyang Technological University, Nanyang, Singapore. [22]Universiti Brunei Darussalam, Gadong, Brunei Darussalam. [23]Program Studi Biologi, Fakultas Teknik, Universitas Samudra, Langsa, Aceh, Indonesia. [24]Tropical Biosphere Research Center, University of the Ryukyus, Okinawa, Japan. [25]National Herbarium, Department of National Botanic Gardens, Peradeniya, Sri Lanka. [26]Sabah Parks, Kota Kinabalu, Sabah, Malaysia. [27]Department of Ecology and Evolutionary Biology, University of Michigan, Ann Arbor, MI, USA. [28]Faculty of Biology, Universitas Jenderal Soedirman, Puwokerto, Indonesia. [29]Faculty of Applied Sciences and Technology, Universiti Tun Hussein Onn Malaysia, Panchor, Johor, Malaysia. [30]Institute of Biological Sciences, Faculty of Science, Universiti Malaya, Kuala Lumpur, Malaysia. [31]The Plant Molecular and Cellular Biology Laboratory, Salk Institute for Biological Studies, La Jolla, CA, USA. [32]Present address: School of Biological Sciences, Nanyang Technological University, Singapore, Singapore. [33]These authors contributed equally: Yee Wen Low, Sitaram Rajaraman, Crystal M. Tomlin. ✉e-mail: low_yee_wen@nparks.gov.sg; cl243@buffalo.edu; e.lucas@kew.org; vaalbert@buffalo.edu

