## [Peer Review File · Nature Communications]

Genomic insights into rapid speciation within the world's largest tree genus *Syzygium*Reviewers' Comments:

Reviewer #1:

Remarks to the Author:

Key results

The authors carried out a de novo chromosome-level assembly of a reference genome for *Syzygium grande* (Myrtaceae), re-examined the paleopolyploidy across the genus and family, assembled >280 *Syzygium* genomes from ~70+X coverage Illumina sequencing data, and then used these assemblies for phylogenetic and network analyses based on (1) genome-wide SNPs and (2) purportedly single-copy nuclear genes (also chloroplast sequences). They also performed population genomics analyses of admixture and clustering and finally examined geography and the distribution of a few morphological characters in light of the phylogeny. Key results: Their findings support the previously established taxonomy of *Syzygium* (divided into 5 subgenera) with improved resolution within one large subgenus. They also conclude that the radiation was rapid and involved significant incomplete lineage sorting (but admixture is not tested appropriately) and possibly clinal fixation of alleles.

Validity

The interpretation of Patterson's statistics appears to be incorrect. Please see: https://compvar-workshop.readthedocs.io/en/latest/contents/03_f3stats/f3stats.html. Further, the presentation of the methods and results is incomplete. No outgroup is stated, no p-values are indicated, both positive and negative z-scores are presented (inconsistent with the conclusion), and the results of the FDR correction are not presented. Tables S4 & S6 are identical, and there is no Table S5; there is no heatmap.

Also, the abstract infers that assessing the importance of adaptation is a key goal of this work. The relevant finding is stated as: "However, the local phylogenetic clustering seems consistent with in situ ecological speciation following simple allopatric lineage splits." Without an accounting of temporal scale, I don't see how sympatry of close relatives is fundamentally different from that of distant relatives. Species ranges shift post-speciation, and this could happen quickly. Thus, co-occurrence of close relatives is not by itself sufficient evidence of ecological speciation (i.e., speciation involving adaptation).

Significance

The significance of this paper is 1) in the tremendous amount of data generated, including hundreds of de novo genome assemblies with high BUSCO scores, and 2) the conclusion of a family-wide paleopolyploidization event. Unfortunately, the paper lacks coherence from the Abstract/Introduction through to the Conclusions and includes problems of interpretation. For example, the Abstract implies a focus on the role of adaptation in species radiations, the Introduction is nearly entirely *Syzygium*-specific and provides no justification for the analyses to be performed, and the stated objectives only partially align with the analyses done. The Results and Discussion are again *Syzygium*-specific. A broader context of interest to a general audience is not presented.

Data and methodology

Processing of the sequencing data and assembly of the genomes appear to be done well, though some detail on the Dovetail HiC-based scaffolding would be helpful.

Analytical approach

Much of the methods section is technically detailed; however, some needed details are missing (see above and below). I don't see any problems with the tests that I am able to evaluate (except for the f3 test above); however, I think additional analyses could be done to strengthen the conclusions (see below).

Suggested improvements

Some of the conclusions of the paper are based on incomplete evidence, and this could be addressed through additional analyses.

For example, the authors point out that the short branch lengths observed within the phylogeny result from some unknown combination of short time and N_e , but then infer the former (i.e., rapid radiation). Demographic modeling with single individuals representing taxa (references below) could be used to more rigorously estimate split times and ancestral population sizes in the subclades with short branch length and thus more rigorously support (or not) the conclusion of rapid radiation. I believe that relationships within subclades are sufficiently close for these approaches. These approaches would also allow another test of admixture (in addition to Patterson's D statistics) – i.e., by adding migration bands to the models and seeing if estimates of divergence times are altered. Practically speaking, one or more of these analyses could be done on a manageable (select) subset of the data.

Gronau, M. J. Hubisz, B. Gulko, C. G. Danko, A. Siepel, Bayesian inference of ancient human demography from individual genome sequences. *Nat. Genet.* 43, 1031–1034 (2011).
S. Schiffels, R. Durbin, Inferring human population size and separation history from multiple genome sequences. *Nature Genetics* 46, 919–925 (2014).
S. Schiffels, K. Wang, "MSMC and MSMC2: The Multiple Sequentially Markovian Coalescent" in *Statistical Population Genomics, Methods in Molecular Biology.*, J. Y. Dutheil, Ed. (Springer US, 2020), pp. 147–166.

Further, the final analyses of the distributions of select morphological traits across the genus-wide phylogeny to draw associations between particular character states and increased diversification do not include any tests of significance. It seems a randomization-based test could be applied here to allow more robust conclusions.

Clarity and context

Again, the paper lacks coherence and a broader context of interest to a non-specialist audience. Aside from evolutionary relationships, the key findings center on paleopolyploidy, diversification rates, dispersal patterns, and possibly morphological traits associated with diversification. All of these objectives/findings should be placed within a broader context. What outstanding hypotheses can be addressed with these data?

Further, where there are more general statements on the evolutionary context of this work, they are often vague or inaccurate. e.g., "Species radiations have long fascinated biologists, but the contribution of adaptation to observed diversity and speciation is still an open question." Or "Species radiations - wherein perplexing amounts of diversity appear to have formed extremely rapidly" or "many *Syzygium* species, particularly within *Syzygium* subg. *Syzygium*, likely branched from one another in rapid succession, yielding true radiations of morphological and ecological diversity." Many statements of objectives should be reworded with more precise language for clarity; e.g., "*Genome structure of Syzygium and its phylogenetic context among angiosperms*" The meaning in italics is unclear.

Regarding justification for targeting *Syzygium* for this work, this statement appears to do this: "However, geographical variation in genus richness is much less pronounced, and therefore understanding diversification within *Syzygium* helps explain large-scale patterns of diversity." – however, the meaning of the statement is not clear.

Generally, many paragraphs in both the Results/Discussion and the Methods are missing statements to clarify why the individual analyses were done. Such statements or opening phrases are required (more condensed in the R&D section) especially for a broad audience. In some cases, the information provided is insufficient to allow evaluation. For example, on page 11, the authors state "A simple explanation for these linear gradations is that allelic variation in *Syzygium* became fixed in consecutive

speciation events, along an ongoing cladogenetic process.” This conclusion is based on “Projections to main principal components.” I don’t know what this means, and the description in the Methods is insufficient. Further, this conclusion should be evident in Fig. 3. However, I had to stare at this figure for a long time to make sense of it. The figure legend needs editing for clarity and accuracy. Ultimately, this analysis is a key feature of the paper, but it needs further justification and clarification.

Otherwise, the Results and Discussion are written in rather technical language and not as accessible to a broad audience as they should be for this journal. I recommend replacing generic subtitles such as: “Phylogeny and population genomics” with statements indicating either the objectives or findings of the analyses.

References

Please see comments above. The Introduction – after the first paragraph – is *Syzygium*-specific, rather than citing the literature needed to justify the study’s objectives. Also, I see that Choi et al 2021 is cited incorrectly (both the island and the figure).

Your expertise

I am familiar with the phylogenetic and population genomics analyses used, but cannot comment on the appropriateness of the specific parameter settings (flags) used. I am not familiar with the analyses used to assess paleopolyploidization or with the Mesquite program.

Reviewer #2:

Remarks to the Author:

The present work represents an attempt to better understand the contribution of adaptation to observed diversity and speciation using the clove genus, *Syzygium*. To do this the authors sequenced, assembled and annotated a reference genome (*Syzygium grandis*) and skim sequenced /annotated an additional ~300 species. Phylogenetic and molecular evolution studies determined the existence of a WGD event shared among Myrtales, and that this event likely occurred that occurred at the base of the order. SNP sets mined from alignment of the skim sequence to the reference, and BUSCO gene sequences enabled construction of species trees of *Syzygium*. The present analysis enabled the establishment/confirmation of major clade relationships and that many *Syzygium* species branched from one another in rapid succession, yielding true radiations of morphological and ecological diversity. This is a well written and comprehensive paper that I believe will have broad interest to the Life Sciences, and to evolutionary and plant biologist in specific.

Given the large number of known species making up the genus *Syzygium*, why was *Syzygium grandis* chosen as the species used to construct the WG reference assembly for the genus?

The authors point out that they examined several Kmer values during transcriptome assembly with TransAbyss (1, 61, 71, 81, 91, 101) but were satisfied that Kmer=25 was adequate for Trinity – why?

It would be helpful for the authors to disclose the various weights applied to each input for evidence modeler and add this to the supplement.

The author’s make an important point regarding the placement of a shared WGD at the base of the order Myrtales. This clears up previous speculation based on 1KP data that WGDs impacting the Lythraceae and Myrtaceae might be independent events. Presumably you can identify signatures of WGD within the fragmented assemblies of the 292 *Syzygium* individuals and outgroups that further confirm this finding? Are there clear examples paralogy throughout the gene sets?

The authors mention the identification of 1,867,173 variants across all 292 samples. Because the SNPs were identified by aligning short read sequences obtained from multiple *Syzygium* accessions to the *S. grandis* reference, the authors also mention that these SNPs are likely from relatively conserved regions of the genome. Because of this, the authors also construct gene trees from single copy BUSCO genes and use ASTRAL to generate a complementary phylogenetic species tree estimation. Is this not a circular confirmation? The universality of the BUSCO gene set implies high conservation – likely also at the sequence level. Would you not expect trees based on BUSCO to confirm trees based on SNPs from region of the genome that are well conserved and shared across species? Perhaps many of the SNPs are within (or near) BUSCO loci. Based on the distribution of the SNPs across the reference genome, and their proximity to genes etc. could the investigators demonstrate that the SNPs are well distributed, estimate the number of genes that these SNPs are within (or associated with), and demonstrate that the confirmation of phylogeny is meaningful?

Reviewer #3:

Remarks to the Author:

Dear Yee Wen and co-authors,

I think this is a well written paper. I could find hardly any faults with the spelling and the grammar. There is nothing wrong with the methods and the results too. What I am wondering though is why this manuscript was deemed to be suitable for Nature Communications.

I think the novel part of the paper is the publishing of the whole genome of *Syzygium grande*? Is that correct? The authors sampled 15% of *Syzygium* species with NGS methods but that's really not that much by today's standards. Previous *Syzygium* studies have been made using Sanger sequencing and sampled as many or more *Syzygium* species than the current one. Similar results were found too. If you disagree and think that your systematic results are new then that needs to be exemplified and highlighted more in the manuscript.

On a similar note, previous Myrtaceae biogeographic studies have shown that the diversification of *Syzygium* is all relatively recent, which is what this paper also shows using a phylogeny based on more genetic information and more modern biogeographic methods. Are your biogeographic results a novel finding though or just reaffirming what we already know? Just like the systematic results you need to highlight what is new about your results.

I think this is a good paper, and you have done lots of work, but I am currently not sure if it is really presenting anything new. My understanding of the premise of Nature Communications is that it publishes papers that are of novel and unique findings. As the manuscript currently stands, I don't see how it could be considered that. I think the manuscript content is probably better suited to MPE or Journal of Biogeography or even Taxon.

We thank the reviewers and editor for their constructive comments. Please refer to our description below of all revisions made, interspersed in blue among the referees' comments. All substantial text revisions in the main text and supplement are highlighted there in red.

Reviewer #1 (Remarks to the Author):

Key results

The authors carried out a de novo chromosome-level assembly of a reference genome for *Syzygium grande* (Myrtaceae), re-examined the paleopolyploidy across the genus and family, assembled >280 *Syzygium* genomes from ~70+X coverage Illumina sequencing data, and then used these assemblies for phylogenetic and network analyses based on (1) genome-wide SNPs and (2) purportedly single-copy nuclear genes (also chloroplast sequences). They also performed population genomics analyses of admixture and clustering and finally examined geography and the distribution of a few morphological characters in light of the phylogeny. Key results: Their findings support the previously established taxonomy of *Syzygium* (divided into 5 subgenera) with improved resolution within one large subgenus. They also conclude that the radiation was rapid and involved significant incomplete lineage sorting (but admixture is not tested appropriately) and possibly clinal fixation of alleles.

Validity

The interpretation of Patterson's statistics appears to be incorrect. Please see: https://compvar-workshop.readthedocs.io/en/latest/contents/03_f3stats/f3stats.html. Further, the presentation of the methods and results is incomplete. No outgroup is stated, no p-values are indicated, both positive and negative z-scores are presented (inconsistent with the conclusion), and the results of the FDR correction are not presented. Tables S4 & S6 are identical, and there is no Table S5; there is no heatmap.

We thank the reviewer for drawing attention to the need for greater description of our use of f_3 statistics, and for the request to describe multiple test correction. We have provided the latter now in a revised Supplementary Data S7, no longer as a single XLS, but instead individual CSV files for each target individual. There are as such no longer any repeated tables. Supplementary Table S5 is now entirely new material; see under our description of new demographic analyses, below. We also provide heatmaps appropriate for our intended points; please see our detailed description of these below.

On the first point, f_3 statistics are highly versatile tools, and we calculate f_3 across all possible triplets to take advantage of this. Measuring allele sharing asymmetries across all three-way combinations of source1, source2, and target individuals/populations permits exploration of different scenarios to explain such sharing including both admixture and drift by descent between source/target. The elegance of this "all-triplets" approach is that no *true* outgroup specification (which would require a 4th taxon, and therefore become f_4 – an ABBA/BABA measure) is required, hence making no pre-assumption of polarity.

As is well known in standard phylogenetics, a three-taxon graph is not a phylogenetic tree. However, allele-sharing asymmetries among the three taxa are still calculable. Positive or negative f_3 values (and their Z-scores) can reflect either close relationships or admixture depending on the format of 3-way comparisons and their allelic interrelationships. Positive f_3 results are usually interpreted as “outgroup- f_3 ” analyses to test for closest relatives of an unknown individual, while negative scores are typically interpreted in terms of admixture. However, negative f_3 scores also appear in the so-called “outgroup case” of hidden admixture (Patterson et al., 2012), or, as we discuss below, when extremely close relationships affected by identity by descent are the cause.

A mathematical proof for negative f_3 values stemming from close relationship and identity by descent (instead of by admixture) is provided at the end of this response; it is formally published on bioRxiv elsewhere (<https://www.biorxiv.org/content/10.1101/2021.12.11.472228v2>).

Again, the present reviewer draws attention to the specification of an “outgroup”, which is in fact specified as part of outgroup f_3 . Patterson’s negatively-scoring outgroup case of hidden admixture should not be confused with outgroup f_3 as developed by Raghavan et al. (2014), where the outgroup reference means the target is *fixed* as an “outgroup” (reference individual/population) – without any addition of a fourth (further outgroup) taxon – to test which source1 is closest to a *fixed* source2. Here, f_3 is calculated $f_3(O;A,B)$, where O is a pre-defined “outgroup”, and either A or B is fixed. The more positive the score, the closer source1 is to fixed source2.

For an example of outgroup f_3 , as shown in Yang et al. 2017 ([https://www.cell.com/current-biology/pdf/S0960-9822\(17\)31195-8.pdf](https://www.cell.com/current-biology/pdf/S0960-9822(17)31195-8.pdf)) to discover the closest relative of a Tianyuan Cave ancient human genome, the investigators fixed Mbuti human as outgroup in every triplet, and then compared various populations A (present day Asians vs. Europeans; noted X below) to assess the closest relative to the Tianyuan Cave human, B. These authors then present the following “heat map” (using our reviewer’s terminology) to display closest relationships to the ancient human.

Figure 1. f_3 (Tianyuan, X; Mbuti) for All Sites Where X is a Present-Day Human Population or an Ancient Individual
 The f_3 statistic ranges from 0.04 to 0.25. A higher value (red) indicates higher shared genetic drift between the Tianyuan individual and the (A) present-day population or (B) ancient individual. The intersection of the dotted lines indicates where the Tianyuan Cave is located. See also Table S2A.

In our present context, we were not interested in performing outgroup f_3 tests, but instead to test for admixture or close relationship via significantly negative f_3 among all possible three-way comparisons. The text is clarified in this respect (see main text, lines 411-418).

In keeping with our descriptions above, and to address the reviewer's comments in the revised manuscript, we now include a different form of f_3 heatmap, one that shows Z-scores for all f_3 triplets corrected for multiple testing using the Benjamini-Hochberg approach. This approach was developed and described in the silver birch genome paper (Salojärvi et al. 2017; <https://www.nature.com/articles/ng.3862>). As already presented in our first submission, these scores are for taxon targets among the “*Syzygium grande* group” (which we also refer to as the NeighborNet “fan”). The heatmaps include all three-way comparisons among a target and two sources, displayed as mirror images across the diagonal, with positive values in orange shades, while negative values are purple. As discussed in our first submission, the results – now depicted in these heatmaps – show no evidence for admixture, but instead signals of close relationship.

As one example, in the heatmap for *Syzygium banksii* (immediately below), all f_3 values are positive (orange scale), which reflects no patterns of close enough allele sharing, either via admixture or descent, among any of the sources (other taxa in the fan) to generate negative values with *S. banksii* as target. The lightest-orange squares showing 2-way comparisons of sources to *S. banksii* are those taxa closest related to it in the test set. The darker orange colors represent more distant relatives.

As another example, shown in the next heatmap, one can see extremely close allele sharing among *Syzygium buxifolium* accessions, particularly between the target buxifolium_SYZ43 and either of both sources buxifolium_SYZ45 and buxifolium_SYZ46 (the dark purple “cross”):

The dark purple (and complete, all-sources) “cross” demonstrates strongly negative Z-scores for both of the two *S. buxifolium* taxa as source1 taxa against ALL other individuals as source2, indicating that these two *S. buxifolium* accessions are extremely closely related to *buxifolium_SYZ43*, i.e., showing highly significant allele sharing through phylogenetic drift (reflected as well by their conspecific status). Note that this illustrates close relationship, with no indication of admixture among these taxa, since ALL source2 individuals demonstrate the allele sharing caused by phylogenetic drift. If there were a smaller sector of purple squares in the fingerprint, one that did not involve all source2 taxa, this could then be interpreted as evidence for admixture. We never see any case of this in our f_3 calculations, as stated in our first submission and the accompanying revision.

To illustrate similar examples where allele sharing patterns from f_3 analyses yield evidence for (1) no close allele sharing vs. (2) close relationship or (3) admixture, we refer the reviewer here to an in-review preprint on bear interrelationships coauthored by 4 of us, from which the proof at the end of this response was also drawn:

Tianying Lan, Kalle Leppälä, Crystal Tomlin, Sandra L. Talbot, George K. Sage, Sean Farley, Richard T. Shideler, Lutz Bachmann, Øystein Wiig, Victor A. Albert, Jarkko Salojärvi, Thomas Mailand, Daniela I. Drautz-Moses, Stephan C. Schuster, Luis Herrera-Estrella, Charlotte Lindqvist. Insights into bear evolution from a Pleistocene polar bear genome. bioRxiv. <https://www.biorxiv.org/content/10.1101/2021.12.11.472228v2>

In the first example, where black bear is the target, various brown and polar bears as source1 and source2 demonstrate only positive f_3 values, and therefore no particularly close patterns of allele-sharing. This result is similar to our example of *Syzygium banksii*, above.

In a second example that shows a similar scenario to our close-relationship demonstration with *Szygyium buxifolium*, using the Alaskan brown bear BB049 as target, individual BB059 demonstrates a complete purple cross of highly significant negative values. BB049 and BB059 are known to be mother and offspring, and thus the allele sharing is purely shared phylogenetic drift from descent, and not admixture. The f_3 “fingerprint” below further helps display closeness among the various sources and target; the light-orange colored complete cross illustrates individuals most closely related to BB049, whereas the dark orange square of values to lower left shows those source1 and source2 individuals farthest away from the target in the SNP set.

In a final example, we refer to the sort of f_3 fingerprint that would be suggestive of admixture instead. Here, polar bear target individual AK034 shows a square sector of negative Z-scores when source1 is either black (BLK) or a brown bear (to BGI-ABC05) and source2 is a non-Alaskan polar bear (from BGI-072 to PB17). Note that there is also a light purple cross of close relationships to AK034 indicated when other AK polar bears are source1 or source2. Similarly to the case of black bear (BLK; above) the square of highly positive (orange) values when any brown or black bear are source1 and source2 to polar bear AK034 shows no close allele sharing whatsoever.

To repeat from above, in the present work we discovered no f_3 fingerprint patterns that would suggest admixture among our *Syzygium* accessions.

Also, the abstract infers that assessing the importance of adaptation is a key goal of this work. The relevant finding is stated as: “However, the local phylogenetic clustering seems consistent with in situ ecological speciation following simple allopatric lineage splits.” Without an accounting of temporal scale, I don’t see how sympatry of close relatives is fundamentally different from that of distant relatives. Species ranges shift post-speciation, and this could happen quickly. Thus, co-occurrence of close relatives is not by itself sufficient evidence of ecological speciation (i.e., speciation involving adaptation).

We thank the reviewer for pointing out that temporal scaling of species-level (not only clade-level) splits was not readily visible in our initial submission. Although only clade-level split times were discussed in our first submission, the time tree (a dated ultrametric tree) that we included indeed includes estimated dates for every taxon split (Supplementary Fig. 20 and Supplementary Data S9). Therefore, we are able to address time scales for sympatric, closely-versus distantly-related *Syzygium* species. Indeed, review of split times for species clusters that are *presently* sympatric within the Danum Valley and Bukit Timah sites reveals that many such splits are indeed old enough to represent taxa that originally speciated geographically, only later to occupy the Danum or Bukit Timah locales. Moreover, some of what appear to be plot-specific phylogenetic clusters may instead be geographic sampling artifacts (in large part, perhaps, because only 15% of species of *Syzygium* were investigated here). Some clusters of mainly Danum or Bukit Timah species in fact include clade members from elsewhere; for example, there is a *S. barringtonioides* specimen from Brunei within an otherwise Danum cluster (both nonetheless being Bornean), and the *S. chloranthum-cerasiforme* clade that was largely sampled from Bukit Timah also contains *S. ampullarium*, which is from Borneo.

As such, we agree with the reviewer that we do not show any particular evidence for ecological speciation in sympatry in the two ecological plots. Instead, with Bukit Timah and Danum species

all arrayed clinally in the PCA analysis, geographic speciation (spatial autocorrelation; see our further comment under the reviewer's "Clarity and context" section, below) is still the most logical interpretation.

One finding in Fig. 2 is, however, clear – the Bukit Timah and Danum *Syzygium* floras are assemblages of diverse lineages, and in our revision we refer to the result in this light only.

An important note on our time tree: when addressing the above details, we realized that in our initial submission we had presented an ultrametric tree based on ASTRAL analysis of the BUSCO data; this was an incorrect operation given the non-convertibility of coalescence branch lengths into divergence times, as has been reported on the ASTRAL developer's github page (see: <https://github.com/smirarab/ASTRAL/issues/37>):

"ASTRAL currently does not output branch lengths for terminal branches. That is probably the root of the issue.

If you want an ultrametric tree only for visualization purposes, just add a constant branch length to all terminals. Use your favorite approach to do that. If you found no alternatives, check out this script: <https://github.com/smirarab/global/blob/master/src/mirphyl/utis/add-bl.py>

If you want an ultrametric tree for dating, I don't think there is currently any good way of doing that using ASTRAL branch lengths."

We have corrected this error in our revision, where we now report timings properly based on an ultrametrically converted RAxML SNP tree. This led to only minor changes in split and crown group timings.

See also another approach to dating below under the reviewer's further comments on demography.

Significance

The significance of this paper is 1) in the tremendous amount of data generated, including hundreds of de novo genome assemblies with high BUSCO scores, and 2) the conclusion of a family-wide paleopolyploidization event. Unfortunately, the paper lacks coherence from the Abstract/Introduction through to the Conclusions and includes problems of interpretation. For example, the Abstract implies a focus on the role of adaptation in species radiations, the Introduction is nearly entirely *Syzygium*-specific and provides no justification for the analyses to be performed, and the stated objectives only partially align with the analyses done. The Results and Discussion are again *Syzygium*-specific. A broader context of interest to a general audience is not presented.

We have revisited the presentation of our various findings to better focus the paper, in part by including descriptive subheadings under Results and Discussion. Our revisions, especially in the

Introduction and Abstract, also involve attempts to better convey the material to a general audience. We have highlighted all edits in the revision in red, as requested (please refer to Reviewer 2's "Clarity and context" section below). Please note, nonetheless, that neither Reviewer 2 nor 3 commented negatively on the first submission's clarity or flow.

Data and methodology

Processing of the sequencing data and assembly of the genomes appear to be done well, though some detail on the Dovetail HiC-based scaffolding would be helpful.

Analytical approach

Much of the methods section is technically detailed; however, some needed details are missing (see above and below). I don't see any problems with the tests that I am able to evaluate (except for the f3 test above); however, I think additional analyses could be done to strengthen the conclusions (see below).

Suggested improvements

Some of the conclusions of the paper are based on incomplete evidence, and this could be addressed through additional analyses.

For example, the authors point out that the short branch lengths observed within the phylogeny result from some unknown combination of short time and N_e , but then infer the former (i.e., rapid radiation). Demographic modeling with single individuals representing taxa (references below) could be used to more rigorously estimate split times and ancestral population sizes in the subclades with short branch length and thus more rigorously support (or not) the conclusion of rapid radiation. I believe that relationships within subclades are sufficiently close for these approaches. These approaches would also allow another test of admixture (in addition to Patterson's D statistics) – i.e., by adding migration bands to the models and seeing if estimates of divergence times are altered. Practically speaking, one or more of these analyses could be done on a manageable (select) subset of the data.

Gronau, M. J. Hubisz, B. Gulko, C. G. Danko, A. Siepel, Bayesian inference of ancient human demography from individual genome sequences. *Nat. Genet.* 43, 1031–1034 (2011).

S. Schiffels, R. Durbin, Inferring human population size and separation history from multiple genome sequences. *Nature Genetics* 46, 919–925 (2014).

S. Schiffels, K. Wang, "MSMC and MSMC2: The Multiple Sequentially Markovian Coalescent" in *Statistical Population Genomics, Methods in Molecular Biology.*, J. Y. Dutheil, Ed. (Springer US, 2020), pp. 147–166.

We thank the reviewer for suggesting that we evaluate past demographic patterns among our genomes, and how these might relate to our other findings of rapid radiation within *Syzygium*. Although the reviewer refers to demographic modeling within single individuals, which we agree would be an excellent addition, two of the three references referred to involve methods designed to utilize allele frequencies calculated from *multiple* individuals. Such MSMC (Multiple Sequentially Markovian Coalescent) approaches, which (except in the trivial 2-haplotype case)

use allele frequencies established from multiple members of a population to make demographic inferences, are not appropriate for our sample, since generating population-level allele frequencies would be limited to only a few species where we have sequenced more than one individual (such as *S. buxifolium*) or would otherwise involve sinking what are now classified as separate species into members of single “populations”. Instead, what is appropriate throughout a reasonably closely related set of taxa is comparative PSMC (Pairwise Sequentially Markovian Coalescent) analysis, which uses the two haploid genomes present in each collection of reads for a given diploid individual to estimate past demography, which can then be interspecifically compared. We ran PSMC demographic curves for most *Syzygium* individuals in the closely interrelated *S. grande* group with Illumina reads mapped against each taxon’s own MaSuRCA genome assembly. Some low-quality MaSuRCA assemblies with low contig N50’s and/or low BUSCO scores also led to irrational curves; the few individuals thus represented were not used in our new inferences. We did investigate a second approach for this project, using *S. grande* reference-mapped reads for each species. However, we do not report on these results in our revision as they were clearly biased by taxon phylogenetic distance to the reference species – a phenomenon encountered in the literature (e.g., <https://onlinelibrary.wiley.com/doi/10.1111/1755-0998.13457>) and one we had already thoroughly investigated for other projects.

In other projects by some of the present authors (either published or in-review), PSMC curves have been successfully used for inferences of population or species splits.

For example, as reported in the Amborella Genome Project (<https://www.science.org/doi/10.1126/science.1241089>), PSMC curves were calculated for 14 individuals of the species, *A. trichopoda*, and used for inferences of population size and timing of evolutionary events that occurred among them.

The distinct ancient coalescences (to right) were interpreted as consistent with the hypothesis that at least two distinct *Amborella* sublineages with different levels of genetic diversity existed

that later converged by 800,000 years ago, followed by admixture and a subsequent bottleneck event between 300,000 and 400,000 years ago.

In another example, referring back to the in-review bear genomics research described under our f_3 fingerprint results above, PSMC was used to establish demographic curves for different bear species and individuals in order to ascertain their diversification patterns over time, including to estimate rough split times.

AK034 is again a polar bear individual, APB is an ancient polar bear genome, and the remainder are brown bear individuals. The SNP variation in all genomes coalesces 3-8 million years ago in ancient time, which the investigators interpreted as the stem lineage subtending these species. The brown bear and polar bear curves (and therefore, their lineages) then start to diverge by 1 million years ago, a split time consistent with another, non-demographic dating method the authors employed.

In our present work on *Syzygium*, plotting together PSMC curves (for all taxa with appropriate MaSurCA assembly qualities) from the *S. grande* group proved very interesting (each curve is colored by their clade/group membership; see subgenus *Syzygium* sub-tree and relevant portion of PCA from main text Fig. 3 below, where the *S. grande* group is to the right in both tree and PCA).

One can see that all taxa coalesce anciently (extreme right among the demographic curves shown below), followed by a peak in effective population size (N_e), various N_e fluctuations/crashes in intermediate times, followed by strong N_e collapse in recent-most time (extreme left). The differences in ancient coalescence time may be attributable to real generation time differences among the taxa, or to differences in past heterozygosity levels (see, e.g., Supplementary note II, <https://www.nature.com/articles/s41588-021-00971-3#Sec56>). For lychee, *ad hoc* shifting of generation times for two populations of different inbreeding/outcrossing profiles (and their resultant heterozygosities) aligned their demographic curves.

For our exercise, we established approximate time (x-axis) and N_e (y-axis) scales by setting generation times (g) to 5 years and mutation rates (μ) to $1E^{-08}$. The curves, regardless, are constants of each analysis and are scalable using any particular g or μ assumption. We have evaluated different g and μ values, and choices of g between 2-12 years (as reported in *Syzygium* literature; see Supplementary Table S5) and μ between 0.8 and 1.6E-08 (well within general plant mutation rate estimates in the literature; see Salojarvi et al., 2019 [<https://www.nature.com/articles/ng.3862>], where the rate estimated for peach [7.77×10^{-9}] was employed for the silver birch genome, and a broad range of rates from 2.9×10^{-9} to 8.9×10^{-8} were evaluated) yield rational x-axis scaling with regard to possible events described

immediately below and similarly reflected in our SNP-based ultrametric time tree, referred to above.

Different groups, e.g., the paraphyletic blue group and the embedded pink lineage, show separation from ancient times going forward in time on the joint graph. The ancient, joint coalescence of variation in these genomes occurs in the range of 10 or so million years ago (using the parameter values reported above), with first N_e decrease happening by between 3 and 1 million years ago, followed by N_e rebounds/changes in the next hundreds of thousands of years, and then final population crashes between ~100-10 thousand years ago. From our SNP-based time tree, the split between the *Syzygium grande* group (and outgroup clades) with their sister lineages in *Syzygium* subgenus *Syzygium* dates to ~2 million years ago, so the ancient PSMC coalescences above could be shared with taxa splitting even deeper in *Syzygium*. Our time tree shows the *S. grande* group's stem lineage splitting ~165,000 years ago, consistent with the lineage splits and N_e rebounds/changes discussed above. In the case of the *S. grande* group, we interpret the joint blue/pink curve tracking in the plots above as membership within a stem lineage that may have already begun segregating. Regarding the latter inference, note how the pink-clade curves are typically lower in N_e than blue-group curves. Then, we interpret the N_e fluctuations closer to present to represent the period of rapid cladogenesis reflected in the "fan-like" reticulate base of the *S. grande* group that is visible in the NeighborNet result. Dates for these events (~100-10 thousand years) are consistent in our ultrametric time tree. Resolving rapid splits like these will be particularly confounded by ILS, which will itself be exacerbated by any N_e size increases (by coalescent theory). Then we interpret the final N_e crashes to the left, closest to present, as the individuation of the clades visible past the stage of the NeighborNet fan (i.e., the "tips " extending from the web in the fan).

We now include these PSMC analyses in a new figure, Fig. 4, and provide relevant materials and methods (lines 879-898) and discussion (lines 488-519) in the main text.

We note that detailed analyses of PSMC curve differences in terms of paleo-ecology/-environment/-geological events are open research questions beyond the scope of the present paper.

Further, the final analyses of the distributions of select morphological traits across the genus-wide phylogeny to draw associations between particular character states and increased diversification do not include any tests of significance. It seems a randomization-based test could be applied here to allow more robust conclusions.

We appreciate that statistical significance of trait associations would be desirable outcomes, but we argue that such a quantitative approach is beyond the scope of the current paper, mostly given that our data contain too many missing observations (? marks; mostly due to incomplete herbarium specimens) as well as polymorphic codings (e.g., red/green/white) – with the profound limitation that almost all current implementations of testing trait correlations require complete data representation and *non*-polymorphic codings. We strongly believe that our qualitative ancestral state reconstructions (Mesquite-based parsimony optimization being

very commonly employed in the literature) are interpretable to the level intended, in this, only one aspect of a much larger work. We have modified language in the manuscript to clarify that these results are preliminary and qualitative only (see lines 585-613).

Clarity and context

Again, the paper lacks coherence and a broader context of interest to a non-specialist audience. Aside from evolutionary relationships, the key findings center on paleopolyploidy, diversification rates, dispersal patterns, and possibly morphological traits associated with diversification. All of these objectives/findings should be placed within a broader context. What outstanding hypotheses can be addressed with these data?

We have considerably revised the introduction in accord with these suggestions; please refer to red-highlighted text there for various substantive changes.

Further, where there are more general statements on the evolutionary context of this work, they are often vague or inaccurate. e.g., “Species radiations have long fascinated biologists, but the contribution of adaptation to observed diversity and speciation is still an open question.” Or “Species radiations - wherein perplexing amounts of diversity appear to have formed extremely rapidly” or “many *Syzygium* species, particularly within *Syzygium* subg. *Syzygium*, likely branched from one another in rapid succession, yielding true radiations of morphological and ecological diversity.” Many statements of objectives should be reworded with more precise language for clarity; e.g., “Genome structure of *Syzygium* and its phylogenetic context among angiosperms” The meaning in italics is unclear.

Regarding justification for targeting *Syzygium* for this work, this statement appears to do this: “However, geographical variation in genus richness is much less pronounced, and therefore understanding diversification within *Syzygium* helps explain large-scale patterns of diversity.” – however, the meaning of the statement is not clear.

Again, we have considerably revised our presentation in accord with these helpful suggestions; please refer to red-highlighted text in our introduction for substantive changes made.

Generally, many paragraphs in both the Results/Discussion and the Methods are missing statements to clarify why the individual analyses were done. Such statements or opening phrases are required (more condensed in the R&D section) especially for a broad audience. In some cases, the information provided is insufficient to allow evaluation.

We have added descriptive subheadings throughout the main text that we believe serve the purpose the reviewer suggests here.

For example, on page 11, the authors state “A simple explanation for these linear gradations is that allelic variation in *Syzygium* became fixed in consecutive speciation events, along an ongoing cladogenetic process.” This conclusion is based on “Projections to main principal components.” I don’t know what this means, and the description in the Methods is insufficient.

Further, this conclusion should be evident in Fig. 3. However, I had to stare at this figure for a long time to make sense of it. The figure legend needs editing for clarity and accuracy. Ultimately, this analysis is a key feature of the paper, but it needs further justification and clarification.

We thank the reviewer for pointing out our earlier in clarity; we have polished our points in revisions that can be seen on main text lines 441-460.

Otherwise, the Results and Discussion are written in rather technical language and not as accessible to a broad audience as they should be for this journal. I recommend replacing generic subtitles such as: “Phylogeny and population genomics” with statements indicating either the objectives or findings of the analyses.

As above, we have added descriptive subheadings that we believe serve the purpose the reviewer suggests.

References

Please see comments above. The Introduction – after the first paragraph – is *Syzygium*-specific, rather than citing the literature needed to justify the study’s objectives. Also, I see that Choi et al 2021 is cited incorrectly (both the island and the figure).

We did in fact incorrectly cite the figure panel (it should have been Fig. 1C), but the island we referred to was intentional, as the Big Island (Hawai’i) shows the sort of PCA cline we draw attention to.

Figure 1C from Choi et al. is shown above (2021);

<https://www.pnas.org/content/118/37/e2023801118>). Please note the linear cline of circles (sample from Hawai'i island) arrayed horizontally along PC1 near PC2 of 0.0. It is true that other island-wise samplings also show clinal patterning (e.g., O'ahu and Kaua'i), but they are not as extensive along a given PC axis.

Your expertise

I am familiar with the phylogenetic and population genomics analyses used, but cannot comment on the appropriateness of the specific parameter settings (flags) used. I am not familiar with the analyses used to assess paleopolyploidization or with the Mesquite program.

Reviewer #2 (Remarks to the Author):

The present work represents an attempt to better understand the contribution of adaptation to observed diversity and speciation using the clove genus, *Syzygium*. To do this the authors sequenced, assembled and annotated a reference genome (*Syzygium grandis*) and skim sequenced /annotated an additional ~300 species. Phylogenetic and molecular evolution studies determined the existence of a WGD event shared among Myrtales, and that this event likely occurred that occurred at the base of the order. SNP sets mined from alignment of the skim sequence to the reference, and BUSCO gene sequences enabled construction of species trees of *Syzygium*. The present analysis enabled the establishment/confirmation of major clade relationships and that many *Syzygium* species branched from one another in rapid succession, yielding true radiations of morphological and ecological diversity. This is a well written and comprehensive paper that I believe will have broad interest to the Life Sciences, and to evolutionary and plant biologist in specific.

Given the large number of known species making up the genus *Syzygium*, why was *Syzygium grandis* chosen as the species used to construct the WG reference assembly for the genus?

Syzygium grande was selected as a representative because it is a well-known member of the most diverse, broadly distributed group of species. *Syzygium grande* was also one of the 12 native Singaporean species that we selected for a pilot study wherein this project was initiated. *Syzygium grande* is also a commonly cultivated shade tree planted along streets in Singapore, and has even been used historically for firebreaks.

The authors point out that they examined several Kmer values during transcriptome assembly with TransAbyss (1, 61, 71, 81, 91, 101) but were satisfied that Kmer=25 was adequate for Trinity – why?

We have added this information as well as our response to the second point below to Supplementary Information, as Supplementary Note 3.

We first ran Transabyss with multiple kmer options to reliably assemble the common transcripts and capture the rare transcripts. For the S1 RNASeq library, the average N50 value across all kmers was 1,500 bp, and the average BUSCO score across kmers was 88.5%. The

average number of transcripts of size ≥ 500 bp for each kmer value was 56,400. Regarding Trinity, we could only select kmer values between 25 and 32. Hence, we decided to go with one iteration and used the default kmer value 25. Trinity, with default kmer 25, provided a substantially better N50 value (2,075 bp), a higher BUSCO score of 91.9%, and yielded 62,744 transcripts of size ≥ 500 bp. We combined all the transcripts using EvidentialGene and assessed the contribution of Transabyss and Trinity to the final transcriptome. From the final count of 57,738 high quality transcripts, 13,681 transcripts came from Trinity, clearly indicating that the default kmer setting in Trinity captured more complete (and likely reliable) transcripts compared to most of the higher kmers from Transabyss.

It would be helpful for the authors to disclose the various weights applied to each input for evidence modeler and add this to the supplement.

We provided the least weight to the self-training gene predictor genemark-es at value 3 since the predictions were mostly fragmented. The next highest weight was provided to the homology-based gene predictor GeMoMa at value 4. Higher weights were provided to *ab initio* gene predictor braker with value 6 for RNASeq library S1, 5 for S2 and 5 for S3 based on higher prediction quality compared to genemark-es and GeMoMa. As recommended by the tool author, we provided the maximum weights to evidence from PASA with the alignments receiving value 7 for RNASeq library S1, 8 for S2 and 9 for S3 while the predicted ORFs received value 10 for S1, 11 for S2 and 12 for S3.

The author's make an important point regarding the placement of a shared WGD at the base of the order Myrtales. This clears up previous speculation based on 1KP data that WGDs impacting the Lythraceae and Myrtaceae might be independent events.

We also generated some new FractBias plots that further solidify the Lythraceae/Myrtaceae commonality of the pan-Myrtaceae event; please see new Supplementary Figure S6.

Presumably you can identify signatures of WGD within the fragmented assemblies of the 292 *Syzygium* individuals and outgroups that further confirm this finding? Are there clear examples paralogy throughout the gene sets?

The contiguities of our MaSuRCA assemblies limit structural comparisons, and we have not generated gene annotations for them as part of this paper, in part because we do not have the appropriate transcriptomic resources. Without gene models, we are presently unable to provide paralog determinations or Ks analyses of polyploidy for the individual species. For future research, however, we will generate preliminary gene models using reference proteome data (including that of *S. grande*) to study gene family dynamics and functional evolutionary differences among *Syzygium* species – topics which we argue are beyond the scope of this already broad phylogenomic paper.

Regarding the Pan-Myrtales polyploidy event, if *Syzygium grande* (Myrtaceae), *Eucalyptus* (Myrtaceae) and *Punica* (Lythraceae) all have it (i.e., that it subtends Myrtaceae+Lythraceae), by logical extension, so must all members of the genus *Syzygium*, not only *S. grande*.

However, we may be able to use our BUSCO data to evaluate the possibility of further WGD events that could have occurred *within Syzygium*. BUSCO completeness (C) is measured in terms of single copy BUSCOs identified (S) plus duplicated (D) BUSCO genes found. I.e., $C=S+D$. For example, expressed in percentages, $C=92.8\%$ BUSCO completeness for *S. kalahiense* derives from $S=89.9\%$ of BUSCOs surveyed found as complete and single copy, plus 3% found complete but duplicated. Elevated D scores relative to S scores, when they occur within a high overall BUSCO C score, might reflect relatively recent polyploidy events if genome assembly size and N50 does not suggest partially diploid assemblies. It is also possible that some diploid regions could have been included in our otherwise haploid MaSuRCA assemblies, and these may affect BUSCO duplicate scores as well. Nonetheless, we have pursued analyses of D scores further.

The distribution of D scores above (x-axis is count of our accessions; y-axis is D) shows an obvious break between $D=7.7$ and 6.7 , and another below D of 4.6 .

Of the 50 individuals with the highest BUSCO D scores, only SYZ355_S_malaccense_AUSTRALIA has a total C score ($\sim 59\%$) less than $\sim 80\%$; as such, the 49 other individuals may have relatively reliable gene space assemblies. Other than this taxon, SYZ355_S_malaccense_AUSTRALIA, with

contig N50 only ~2,200 bp, all other 49 individuals range from N50 ~6,000-30,000, and none other than this individual have grossly inflated genome sizes (See Supplementary Table S3).

The species with highest D score, *Syzygium cumini* (with $C84.3 = S71.7 + D12.6$), is a known neopolyploid with double the number of chromosomes ($n=22$) as the number of large scaffolds in our (paleotetraploid) *Syzygium grande* reference ($n=11$).

See:

[http://ccdb.tau.ac.il/Angiosperms/Myrtaceae/Syzygium/Syzygium%20cumini%20\(L.\)%20Keels](http://ccdb.tau.ac.il/Angiosperms/Myrtaceae/Syzygium/Syzygium%20cumini%20(L.)%20Keels) – from Rice et al. 2015. The Chromosome Counts Database (CCDB) – a community resource of plant chromosome numbers. New Phytol. 206(1): 19-26.

As such, we hypothesize that some other species with high D scores (perhaps those between $D = 12.6-7.7$) may also be neopolyploids.

Another clear neopolyploid from the literature is *Syzygium jambos*, with $n=22$ (see: [http://ccdb.tau.ac.il/Angiosperms/Myrtaceae/Syzygium/Syzygium%20jambos%20\(L.\)%20Alston](http://ccdb.tau.ac.il/Angiosperms/Myrtaceae/Syzygium/Syzygium%20jambos%20(L.)%20Alston)). While our *S. jambos* does have a BUSCO D score in the top 50, it is only 3.9%, which may signify that our criterion used here is of limited utility. Moreover, *S. samarangense* has mainly neopolyploid chromosome counts in the literature (see: [http://ccdb.tau.ac.il/Angiosperms/Myrtaceae/Syzygium/Syzygium%20samarangense%20\(Blum](http://ccdb.tau.ac.il/Angiosperms/Myrtaceae/Syzygium/Syzygium%20samarangense%20(Blum)

e)%20Merr.%20&%20L.%20M.%20Perry/), but our sample has a D score of 2.9%. There certainly may also be true “polyploid series” variation within a number of *Syzygium* species.

As such, we chose to point to only the interesting and verified case of *Syzygium cumini* in our revision (please see main text lines 291-296).

The authors mention the identification of 1,867,173 variants across all 292 samples. Because the SNPs were identified by aligning short read sequences obtained from multiple *Syzygium* accessions to the *S. grandis* reference, the authors also mention that these SNPs are likely from relatively conserved regions of the genome. Because of this, the authors also construct gene trees from single copy BUSCO genes and use ASTRAL to generate a complementary phylogenetic species tree estimation. Is this not a circular confirmation? The universality of the BUSCO gene set implies high conservation – likely also at the sequence level. Would you not expect trees based on BUSCO to confirm trees based on SNPs from region of the genome that are well conserved and shared across species? Perhaps many of the SNPs are within (or near) BUSCO loci. Based on the distribution of the SNPs across the reference genome, and their proximity to genes etc. could the investigators demonstrate that the SNPs are well distributed, estimate the number of genes that these SNPs are within (or associated with), and demonstrate that the confirmation of phylogeny is meaningful?

Indeed, we agree with the reviewer that both data sets should contain relatively conservative phylogenetic markers. The SNPs were generated against a single species’ reference genome, and so likely have very mixed coverage among the various individuals due to (1) phylogenetic distance to that reference, and (2) differences in noncoding (evolutionarily labile) DNA content that in part contributes to genome size differences observed among the MaSuRCA assemblies. The BUSCO gene sets, even if some of their base differences overlap with our SNPs, represent a completely independent discovery approach based on the self MaSuRCA assemblies – no reference genome and any potential mapping biases to it were involved. Importantly, the BUSCO approach also permitted us to run a locus-wise coalescent species tree analysis to complement the genome-wide average tree afforded by the SNP variation. We chose not to further investigate SNP distribution or density due to the above-described distinctiveness of the two data forms.

Reviewer #3 (Remarks to the Author):

Dear Yee Wen and co-authors,

I think this is a well written paper. I could find hardly any faults with the spelling and the grammar. There is nothing wrong with the methods and the results too. What I am wondering though is why this manuscript was deemed to be suitable for Nature Communications.

I think the novel part of the paper is the publishing of the whole genome of *Syzygium grande*? Is that correct? The authors sampled 15% of *Syzygium* species with NGS methods but that's really not that much by today's standards. Previous *Syzygium* studies have been made using Sanger

sequencing and sampled as many or more *Syzygium* species than the current one. Similar results were found too. If you disagree and think that your systematic results are new then that needs to be exemplified and highlighted more in the manuscript.

On a similar note, previous Myrtaceae biogeographic studies have shown that the diversification of *Syzygium* is all relatively recent, which is what this paper also shows using a phylogeny based on more genetic information and more modern biogeographic methods. Are your biogeographic results a novel finding though or just reaffirming what we already know? Just like the systematic results you need to highlight what is new about your results.

I think this is a good paper, and you have done lots of work, but I am currently not sure if it is really presenting anything new. My understanding of the premise of Nature Communications is that it publishes papers that are of novel and unique findings. As the manuscript currently stands, I don't see how it could be considered that. I think the manuscript content is probably better suited to MPE or Journal of Biogeography or even Taxon.

We thank the reviewer for their positive response to the general body of our work, but respectfully disagree with the opinion that it might not be unique or novel enough for publication in Nature Communications, and to our knowledge this was not questioned by our other reviewers. Aside from the novelty of our high-quality *Syzygium* grande genome assembly, our findings here have several major points of new general and specific interest:

- (1) We support a single Myrtales-wide WGD event instead of multiple independent events within Myrtales, as presented by the 1KP Nature paper (<https://www.nature.com/articles/s41586-019-1693-2>). This represents an important advance in understanding the role of polyploidy in this large order.
- (2) Regarding our species sampling, most next generation sequencing (NGS) studies in the modern era use either reduced representation (e.g., RADseq) sequencing or Hybseq (bait-based) recovery of a restricted set of marker genes. Here, we have provided 292 complete genome sequences, each sequenced to a depth of 30 Gb, which in most cases was sufficient enough for us to generate decent draft MaSurCA assemblies and genome-wide SNPs for downstream analyses. Both permitted close looks at the effects of incomplete lineage sorting during *Syzygium*'s rapid diversification, and the latter permitted our detailed studies of the possibility of introgression during the evolutionary history of the genus. Furthermore, our demographic analyses new to this revised manuscript version could not have been robust with reduced representation SNP data.
- (3) We robustly resolved relationships within *Syzygium* subgenus *Syzygium* (the most diverse clade in the genus), which had only been poorly understood before from statistically weakly-supported analyses of many fewer phylogenetic markers.
- (4) We uncovered that the species-rich subgenus has undergone rapid radiation and is confounded by incomplete lineage sorting, and not ancient introgression. Such patterns could not have been resolved by the earlier molecular analyses that examined very few plastome and nuclear markers generated through PCR amplification.

Appendix: Proof (by Kalle Leppälä) describing the appearance of negative f_3 scores, on the basis of identity by descent, in cases of extremely close relationship. A version of this text appears in the bioRxiv paper on bears referred to above (<https://www.biorxiv.org/content/10.1101/2021.12.11.472228v2>).

In the case of testing for admixture, Patterson et al. (2012; <https://academic.oup.com/genetics/article/192/3/1065/5935193>) showed that when the f_3 -statistic, $f_3(C; A, B)$, is significantly negative, C can be interpreted as admixed (although a positive value does not necessarily indicate that C is not admixed). The f_3 -statistic captures drift on overlapping paths from C to A and C to B, and if C is admixed, some of this drift can contribute a negative value to the statistics. This can happen either when A or B is an outgroup, while the other is closely related to one of the source populations of the admixture, or when A and B are related to different source populations. The closer A and B are to the source populations of the admixture, the more negative f_3 becomes, so we can search for the most likely source population for an admixed C by looking for the pair (A, B) that gives the most negative f_3 .

It is important to note that when C and either A or B are from the same population and the other is an outgroup, we are also likely to sometimes see a negative f_3 due to recent family structure. To see this, consider an example where A, B and C are single individuals, and A and C come from the same population. To formalize the relationship between A and C, we denote the proportion of the genome where A and C are independent by π_0 , the proportion where they have exactly one chromosome identical by descent (IBD) by π_1 , and the proportion where they have both chromosomes IBD by π_2 . When two chromosomes are not IBD, the alleles can still match by chance (identical by state, IBS), and we model the allelic states using a binomial distribution and the population level frequencies c (for A and C who come from the same population) and b (for B). When an estimator for an f_3 -statistic is constructed from estimators \hat{a} , \hat{b} and \hat{c} for the allele frequencies, a bias correction is necessary as derived by Reich et al. (2009; <https://www.nature.com/articles/nature08365>), written in an equivalent form:

$$\hat{f}_3(C; A, B) = (\hat{c} - \hat{a})(\hat{c} - \hat{b}) - \frac{\hat{c}(1 - \hat{c})}{n_c - 1},$$

where n_c is the sample size of C; in our example $n_c = 2$ for the two chromosomes. In the presence of enough loci with chromosomes IBD, this estimator is no longer unbiased and doesn't match the correct value $f_3(C; A, B) = 0$. We break the analysis of expected behavior of the estimator at a random locus into three cases depending on the number of chromosomes IBD between A and C.

Case 0) All alleles are independent, probability π_0 . By the standard rules for expected value, $E((\hat{c} - \hat{a})(\hat{c} - \hat{b})) = E(\hat{c}^2) - E(\hat{c})E(\hat{a}) - E(\hat{c})E(\hat{b}) + E(\hat{a})E(\hat{b})$. The following table demonstrates the values $E(\hat{c}) = c$ and $E(\hat{c}^2) = 0.5c(c + 1)$:

probability	genotype of C	\hat{c}	\hat{c}^2
-------------	---------------	-----------	-------------

c^2	mm	1	1
$2c(1 - c)$	mM	0.5	0.25
$(1 - c)^2$	MM	0	0
		$E(\hat{c}) = c$	$E(\hat{c}^2) = 0.5c(1 + c)$

Similarly, $E(\hat{a}) = c$ and $E(\hat{b}) = b$, so we get $E((\hat{c} - \hat{a})(\hat{c} - \hat{b})) = 0.5c(1 - c)$.

Case 1) One of the two chromosomes are IBD, probability π_1 . This time, \hat{a} and \hat{c} are no longer independent, and we only have $E((\hat{c} - \hat{a})(\hat{c} - \hat{b})) = E(\hat{c}^2) - E(\hat{c}\hat{a}) - E(\hat{c})E(\hat{b}) + E(\hat{a})E(\hat{b})$. The following table and some arithmetics demonstrate the value $E(\hat{c}\hat{a}) = 0.25c + 0.75c^2$:

probability	genotype of C	genotype of A	\hat{c}	\hat{a}	$\hat{c}\hat{a}$
c^3	mm	mm	1	1	1
$c^2(1 - c)$	mm	mM	1	0.5	0.5
$c^2(1 - c)$	mM	mm	0.5	1	0.5
$c(1 - c)^2$	mM	mM	0.5	0.5	0.25
$c^2(1 - c)$	Mm	Mm	0.5	0.5	0.25
$c(1 - c)^2$	Mm	MM	0.5	0	0
$c(1 - c)^2$	MM	Mm	0	0.5	0
$(1 - c)^3$	MM	MM	0	0	0
					$E(\hat{c}\hat{a}) = 0.25c + 0.75c^2$

We used the bold font weight to mark the genotype that was IBD. The rest of the terms are already familiar to us, so we get $E((\hat{c} - \hat{a})(\hat{c} - \hat{b})) = 0.25c(1 - c)$.

Case 2) Both chromosomes are IBD, probability π_2 . Trivially $E((\hat{c} - \hat{a})(\hat{c} - \hat{b})) = 0$. In all three cases the bias correction gives $E(-\hat{c}(1 - \hat{c})) = -(E(\hat{c}) - E(\hat{c}^2)) = -0.5c(1 - c)$. Noting that $2c(1 - c)$ is the heterozygosity, we have now derived the formula:

$$E(\hat{f}_3(C; A, B)) = \frac{HET_c}{4} \left(\pi_0 + \frac{1}{2}\pi_1 - 1 \right).$$

In the case of a mother and a daughter, such as the brown bears BB049 and BB059 (please see the f_3 fingerprint shown above), we have $\pi_0 = \pi_2 = 0$, $\pi_1 = 1$. Then the estimator of the f_3 -statistic necessarily becomes negative without presence of any admixture at all, as is the case whenever $\pi_0 < 1$.

Reviewers' Comments:

Reviewer #1:

Remarks to the Author:

I commend the authors for their work on revising the manuscript and for their detailed explanations of these revisions. However, the novelty and broader context of the results are still issues. As two reviewers pointed out in the first manuscript (using different words), the paper is *Syzygium*-specific and lacks a broader context of interest to non-*Syzygium* biologists. *Syzygium* is clearly an important tropical tree genus, and I think the inferences on the pattern and timing of the spread of this genus out of Sahul are the most valuable contribution of this work. Otherwise, while insights from the evolution of this species-rich group may be applicable to the evolution of other tropical groups (as implied by the authors), the conclusions/inferences of this work are either: 1) not particularly strong; i.e., paleopolyploidy is confirmed in this group, but its role in generating high species richness remains unclear; both neutral and adaptive processes likely contributed to the *Syzygium* radiation, but the relative roles remain unknown; the findings on morphological transitions (corolla, fruit color, and inflorescence) across the genus are not even included in the abstract; or 2) not particularly novel: i.e., the *Syzygium* radiation was likely rapid and involves incomplete lineage sorting. If the conclusion that the high species richness of tree genus, *Syzygium*, resulted predominantly from neutral processes could be supported more robustly – I think this conclusion would mark a major advance of interest to evolutionary biologists.

This lack of a broader context is seen in the first two sentences of the abstract, which appear to introduce two different problems.

The manuscript still contains text that needs rewording (commented on in the original version): e.g., "Genome structure of *Syzygium* and its phylogenetic context among angiosperms," "yielding 634 true radiations of morphological and ecological diversity"

Editing is incomplete on page 10; lines 394 (we don't provide a formal test) & 411 (we did f3 tests) contradict each other.

Line 467: Edit for clarity.

Reviewer #2:

Remarks to the Author:

Thank you for careful consideration of my comments on your manuscript. I appreciate the care and attention you have taken while addressing my concerns, as well as those of the other reviewers. I feel this is an interesting and comprehensive study that has been described in a well written report.

Reviewer #3:

Remarks to the Author:

I think you have significantly revised and improved the manuscript. The first time that I reviewed your paper I didn't appreciate what your significant findings were. I think that with the revision they are now more obvious.

Response to referees (1st revision):

We thank the editor and referees for their re-review of our manuscript. We address their points as best possible below. Please refer to our description below of revisions made, interspersed in blue among the referees' comments. All substantial text revisions in the main text and supplement are highlighted there in red.

Reviewer #1 (Remarks to the Author):

I commend the authors for their work on revising the manuscript and for their detailed explanations of these revisions.

We thank the reviewer for their comment.

However, the novelty and broader context of the results are still issues. As two reviewers pointed out in the first manuscript (using different words), the paper is *Syzygium*-specific and lacks a broader context of interest to non-*Syzygium* biologists. *Syzygium* is clearly an important tropical tree genus, and I think the inferences on the pattern and timing of the spread of this genus out of Sahul are the most valuable contribution of this work. Otherwise, while insights from the evolution of this species-rich group may be applicable to the evolution of other tropical groups (as implied by the authors), the conclusions/inferences of this work are either: 1) not particularly strong; i.e., paleopolyploidy is confirmed in this group, but its role in generating high species richness remains unclear; both neutral and adaptive processes likely contributed to the *Syzygium* radiation, but the relative roles remain unknown; the findings on morphological transitions (corolla, fruit color, and inflorescence) across the genus are not even included in the abstract; or 2) not particularly novel: i.e., the *Syzygium* radiation was likely rapid and involves incomplete lineage sorting. If the conclusion that the high species richness of tree genus, *Syzygium*, resulted predominantly from neutral processes could be supported more robustly – I think this conclusion would mark a major advance of interest to evolutionary biologists. This lack of a broader context is seen in the first two sentences of the abstract, which appear to introduce two different problems.

We feel we successfully revised the general readability substantially in our first revision, and we believe we have satisfactorily highlighted the various novelties in this revision as well. Regarding the abstract, unfortunately it had to be cut substantially for length at this time, so trait evolutionary inferences were not included.

The manuscript still contains text that needs rewording (commented on in the original version): e.g., “Genome structure of *Syzygium* and its phylogenetic context among angiosperms,”

This heading no longer appears in the manuscript.

“yielding 634 true radiations of morphological and ecological diversity”

We have removed the word “true” in the context noted above.

Editing is incomplete on page 10; lines 394 (we don’t provide a formal test) & 411 (we did f3 tests) contradict each other.

We have removed the first mention of formal testing and modified the sentence as follows, leaving the mention of admixture testing in connection to f3 analysis:

“We therefore propose ILS to be a likely underlying causal factor for some of the K mixtures given both the short coalescence branch lengths on the ASTRAL species tree and the reticulation of the NeighborNet.”

Line 467: Edit for clarity.

We have chosen to maintain our current wording.

Reviewer #2 (Remarks to the Author):

Thank you for careful consideration of my comments on your manuscript. I appreciate the care and attention you have taken while addressing my concerns, as well as those of the other reviewers. I feel this is an interesting and comprehensive study that has been described in a well written report.

We thank the reviewer for their comment.

Reviewer #3 (Remarks to the Author):

I think you have significantly revised and improved the manuscript. The first time that I reviewed your paper I didn't appreciate what your significant findings were. I think that with the revision they are now more obvious.

We thank the reviewer for their comment.